# Design of multi-epitope vaccine candidate against Brucella type IV secretion system (T4SS)

**Zhengwei Yin[1], Min Li[1], Ce Niu[1], Mingkai Yu[2], Xinru Xie[1], Gulishati Haimiti[1], Wenhong Guo[1], Juan Shi[1], Yueyue He[2], Jianbing Ding[3]*, Fengbo Zhang[3,4]**

1 The First Affiliated Hospital of Xinjiang Medical University, Xinjiang, China, 2 Department of Immunology, School of Basic Medical Sciences, Xinjiang Medical University, Xinjiang, China, 3 State Key Laboratory of Pathogenesis, Prevention, Treatment of Central Asian High Incidence Diseases, The First Affiliated Hospital of Xinjiang Medical University, Xinjiang, China, 4 Department of Clinical Laboratory, The First Affiliated Hospital of Xinjiang Medical University, Xinjiang, China

* 765219598@qq.com (FZ); 1601379937@qq.com (JD)

## Abstract

Brucellosis is a common zoonosis, which is caused by Brucella infection, and Brucella often infects livestock, leading to abortion and infertility. At present, human brucellosis remains one of the major public health problems in China. According to previous research, most areas in northwest China, including Xinjiang, Tibet, and other regions, are severely affected by Brucella. Although there are vaccines against animal Brucellosis, the effect is often poor. In addition, there is no corresponding vaccine for human Brucellosis infection. Therefore, a new strategy for early prevention and treatment of Brucella is needed. A multi-epitope vaccine should be developed. In this study, we identified the antigenic epitopes of the Brucella type IV secretion system VirB8 and Virb10 using an immunoinformatics approach, and screened out 2 cytotoxic T lymphocyte (CTL) epitopes, 9 helper T lymphocyte (HTL) epitopes, 6 linear B cell epitopes, and 6 conformational B cell epitopes. These advantageous epitopes are spliced together through different linkers to construct a multi-epitope vaccine. The silico tests showed that the multi-epitope vaccine was non-allergenic and had a strong interaction with TLR4 molecular docking. In immune simulation results, the vaccine construct may be useful in helping brucellosis patients to initiate cellular and humoral immunity. Overall, our findings indicated that the multi-epitope vaccine construct has a high-quality structure and suitable characteristics, which may provide a theoretical basis for the development of a Brucella vaccine.

**Data Availability Statement:** All relevant data are within the paper and its Supporting Information files The data derived from public domain information: Uniprot database (https://www.uniprot.org/) And PDB library (https://www.rcsb.

## 1. Introduction

Brucella is a tiny, globular, gram-negative bacteria that is non-budding, non-flagellated and does not form pods. Brucella can be divided into 12 species and have been described in detail in previous studies [1]. Although Brucella *melitensis* primarily infects sheep and goats, it also frequently causes brucellosis in humans [2]. Brucellosis is one of the most common zoonotic

org/) The data that supports the findings of this study are available in the methods and/or supplementary material of this article For any questions, please contact the corresponding author.

**Funding:** This study was supported by grants (No. 81860352, No.81860375, No.81560322) from the National Natural Science Foundation of China and funds for the Xinjiang Key construction Project of the 13th Five-Year Plan (basic medicine) and Tianshan Youth Talent Program of Xinjiang Uygur Autonomous Region, State Key Laboratory of Pathogenesis, Prevention and Treatment of High Incidence Diseases in Central Asia Fund (SKL-HIDCA2021-JH11, https://caskl.xjmu.edu.cn/ )","Xinjiang Uygur Autonomous Region science and technology support project (2022E02061, http://kjt. xinjiang.gov.cn/kjt/c100870/201111/ acd5ead987f945cba2498ce413a36388.shtml)" and "Youth Science and technology top talent Program (2022TSYCCX0112, http://kjt.xinjiang.gov.cn/kjt/ c100870/201111/ acd5ead987f945cba2498ce413a36388.shtml) ". Author: F.B Zhang. The funders had no role in study design, data collection and analysis, decision to publish, or preparation of the manuscript.

**Competing interests:** The authors have declared that no competing interests exist.

**Abbreviations:** MEV, multi-epitope vaccine; CTL, cytotoxic T lymphocyte; HTL, helper T lymphocyte; LBE, Linear B-cell epitopes; CBE, Conformational B-cell epitopes; GRAVY, grand average of hydropathicity; RMSD, Root mean square deviations; hBD3, human β-defensin-3; TLR4, Toll-like receptor 4; LPS, lipopolysaccharides; CAI, codon adaptation index; PCR, polymerase chain reaction; MCS, multiple cloning site; NMA, normal mode analysis.

infections worldwide [3, 4]. The main clinical manifestations are fever, weakness, arthralgia, and muscle pain [5]. The main routes of transmission are the gastrointestinal tract, skin, mucous membranes, respiratory tract, blood body fluids, and aerosols. For example, direct or indirect contact with infected animals, and consumption of their raw meat or dairy products [6, 7]. In recent years, the incidence of human brucellosis in China has increased dramatically [8]. However, there are still many clinical challenges in the diagnosis of brucellosis because of its non-specified clinical features, the slow growth rate in blood cultures, and the complexity of its serodiagnosis [9]. In addition, the disease is easy to develop into chronic, which can affect multiple organs at the same time. Therefore, developing a new approach to early prevention of brucellosis is necessary [10, 11].

Vaccines are an ideal way to prevent brucellosis. Unfortunately, there is still no available human brucellosis vaccine. All commercially available animal vaccines are based on live attenuated strains of Brucella that can induce abortions in pregnant animals and are potentially infectious to humans. Therefore, An efficient Brucella vaccine is urgently needed [12]. Currently, there are numerous methods being used in vaccine research, including reverse genetically modified live-attenuated vaccines, vector vaccinology, DNA vaccines, and multi-epitope vaccines (MEV). Specially, reverse vaccinology (RV) has proven to be a very effective approach [13, 14]. On the other hand, uses RV approach to vaccine prediction, which can improve its safety and efficacy [15]. The VirB system (VirB1-VirB12) has been shown to be present in all Brucella species and is highly conserved [16]. It disrupts cellular pathways, induces a host immune response by secreting effectors, promotes replication of Brucella in host cells and induces persistent infection [17]. VirB8 is one of the core components of the type IV secretion system (T4SS) and has been shown in previous experiments to be immunogenic and suitable as a candidate protein for vaccine design [18]. VirB10 is an important functional protein of Brucella abortus and one of the core components of the type IV secretion system (T4SS) [19]. VirB10 has been shown to have good immunogenicity in animal models of infection and is a good choice for vaccine design [20]. Ultimately, both proteins are suitable for MEV design against Brucella *melitensis*.

We analyzed the epitopes of VirB8 and VirB10 using various bioinformatics methods such as IEDB, NetCTLPAN1.1, NETMHCIIpan4.0, and ABCpred. Epitope bonding requires a linker. We use AAY, GPPGPG, and KK to link CTL epitopes, HTL epitopes, and B-cell epitopes [21]. However, epitopes linked vaccine peptides are poor immunogens and susceptible to enzymatic degradation, so we chose specific adjuvants to enhance and stabilize the immunogenicity of the vaccine peptides [22]. We analyzed the physicochemical properties, antigenicity and sensitization of MEV and performed a model assessment of the predicted secondary and tertiary structures of MEV. In addition, the vaccination must work appropriately with the host's immunological receptors in order to trigger the immune response from the host. Therefore, molecular docking was used to check the vaccine construct models' ability to bind to host immunological receptors. TLR4 is the main receptor of LPS and can be activated by LPS. Subsequently, the activity of the MEV-TLR4 complex is also activated, and the immune response of the body is initiated [23]. Then, the vaccine construct was codon optimized to ensure its proper translation inside the host. finally, the stability of the complex was validated by a molecular dynamics simulation.

## 2. Materials and methods

### 2.1 Material sources

Find the amino acid sequences of Brucella *melitensis* VirB8(Serial No. Q9RPX7, https://www. uniprot.org/uniprotkb/Q9RPX7/entry) and VirB10 (Serial No. Q8YDZ0, https://www.uniprot.

org/uniprotkb/Q8YDZ0/entry) in the UniProt database (https://www.uniprot.org/). The amino acid sequences in Jalview were compared by MAFFT to analyze their homology [24].

## 2.2 Research methodology

**2.2.1 Selection of target proteins.** ProtParam (http://web.expasy.org/protparam/) software was applied to analyze the physicochemical properties of the proteins and MEVs. This included the number of amino acids, the molecular formula, the instability index, and the overall mean of the water solubility (GRAVY). VaxiJen 2.0 (http://www.ddg-pharmfac.net/vaxijen/VaxiJen/VaxiJen.html) was applied to analyze the antigenicity of the target proteins with a threshold value of 0.4 [25]. The VaxiJen 2.0 server provided alignment-independent predictions of potential antigens based on physicochemical characteristics. The protein sequences were transformed automatic and cross-covariance (ACC) into a uniform vector of the main amino acid properties, with an antigenic threshold of 0.4 for each bacterial protein. For this purpose, we also compared the hydrophilicity and stability of the target protein, and analyzed its allergenicity through the online website AllergenFP v.1.0.

**2.2.2 Prediction of signal peptides.** The prediction of protein signal peptides was implemented by SignalP5.0 and LiPOP1.0 (https://services.healthtech.dtu.dk/service.php?LipoP-1.0). The type of signal peptide predicted is related to SP (Sec/SPI); the cleavage location is represented by CS; and Other is the likelihood that the sequence lacks any sort of signal peptide. Finally, we chose the merged set as the final result [26].

**2.2.3 Prediction of protein T-cell epitopes.** T cell epitopes consist of MHC class I and class II molecules, and CD8 T cells become cytotoxic T lymphocytes (CTL) upon recognition of the CD8 epitope. At the same time, the triggered CD4 T cells become helper T lymphocytes (HTL) or regulatory (Treg) T cells [27, 28]. We selected the Xinjiang High frequency alleles HLA-A*1101(13.46%), HLA-A*0201(12.50%), HLA-A*0301(10.10%), HLA-DRB1*0701 (16.35%), HLA-DRB1*1501(8.65%)and HLA-DRB1*0301(7.69%) to predict CTL and HTL epitopes [29]. CTL epitopes of target proteins are predicted by IEDB (http://tools.immuneepitope.org/) and the NetCTLpan1 server (https://services.healthtech.dtu.dk/service.php?NetCTLpan-1.1) [30, 31]. For CTL epitope prediction, the three alleles of HLA-A were selected with a length of 10 and the other original thresholds were unchanged. As NetCTLpan1.1 starts counting from 0, care should be taken to add 1 to the sequence when comparing the results with those of IEDB at the end. Application of IEDB and NetMHC-IIpan-4.0 (https://services.healthtech.dtu.dk/service.php?NetMHCIIpan-4.0) for prediction of HTL epitopes of target proteins [32]. For HTL epitope prediction, three allele lengths, such as HLA-DRB1, were chosen to be 15. The default thresholds for NetMHC-IIpan-4.0 remain unchanged. At the same time, we conducted allergenicity analysis on T cell epitopes using online analysis software AllergenFP v.1.0. Ultimately, the overlapping sequences of the top 5 with high scores, non-allergenic and high immunogenicity from two software were chosen as T cell dominant epitopes. The population coverage analysis of alleles was selected based on previous research [29].

**2.2.4 Prediction of protein B-cell epitopes.** B-cell epitopes consist of linear and conformational epitopes. ABCpred (https://webs.iiitd.edu.in/raghava/abcpred/ABC_submission.html) was used to predict the selection of dominant B-cell linear epitopes with an overall prediction accuracy of 65.93%. The corresponding sensitivity, specificity, and positive predictive values were 67.14%, 64.71%, and 65.61%, respectively. The prediction of conformational epitopes was performed by Ellipro of IEDB (http://tools.iedb.org/ellipro/)and we selected sequences with high scores. Similarly, B-cell dominant epitopes were also analyzed for allergenicity using online software AllergenFP v.1.0. Finally, the linear and conformational dominant

epitopes of B cells with high scores, high Antigenicity, and non-allergenic were selected for MEV construction.

**2.2.5 Molecular docking of T-cell epitopes to HLA alleles.** Use the HDOCK server for molecular docking. We selected HLA class I (HLA-A*02:01) and HLA class II (HLA-DRB1*01:01) alleles for molecular docking with T-cell epitopes and eventually discovered the interactions between the alleles and T-cell epitopes.

**2.2.6 Vaccine construction for MEV.** We choose non-toxic, non-allergenic, and advantageous table positions. CTL epitopes are linked to AAY linkers and linked to HTL epitopes with GPGPG linkers; HTL epitopes are linked to GPGPG linkers and linked to B-cell epitopes with KK linkers; B-cell epitopes are linked with KK linkers [21]. To increase the antigenicity of MEV, the human β-defensin-3 sequence (sequence no. Q5U7J2)and the PADRE sequence were linked with the help of an EAAAK junction at the N terminus [33]. Finally, a polyhistidine tag is added to the C-terminus to obtain the complete vaccine protein sequence.

**2.2.7 Physicochemical properties, antigenicity, solubility, and allergenicity of MEV.** ProtParam software was applied to analyze the physicochemical properties of MEV. VaxiJen was applied to analyze the antigenicity with a threshold of 0.4 and SOLpro (http://scratch. proteomics.ics.uci.edu/) was applied to predict the solubility of MEV. A two-stage SVM design was adopted based on the numerous representations of the first-order amino acid sequence. In summary, the forecast had an overall accuracy of 74% at a matching probability of 30.5. and a threshold value of 0.5 [34]. Finally, AllergenFP (http://www.ddg-pharmfac.net/AllergenFP/) was applied to analyze its allergenicity.

**2.2.8 Projections for secondary and tertiary structures.** SOPMA online analysis software (http://npsa-pbil.ibcp.fr/cgibin/npsa_automat.pl?page=/NPSA/npsa_sopma.html) was used to forecast and analyze the proportions and distributions of the vaccine build sequence's alpha helix, beta-turn, random coil, and extended strand. Afterward, we applied RoseTTAFold (Submit a job for structure prediction (bakerlab.org)) to predict the tertiary structure of MEV. RoseTTAFold is a "three-track" neural network. In this network architecture, the one-dimensional, two-dimensional, and three-dimensional information of proteins can flow back and forth, exchanging information, allowing the neural network to integrate all information and infer the relationship between the chemical components of proteins and their folding structures. The predicted model has sufficient structural similarity to the real structure to provide a successful MR solution in all cases [35].

**2.2.9 Quality assessment of predictive models.** The quality of the three-level structural model of the MEV was assessed using the SWISS-MODEL structural assessment (https:// swissmodel.expasy.org/assess). The prediction method of SWISS-MODEL can be described as follows: α- Spiral and β- Folding based "rigid segment assembly". The software first finds known proteins that are homologous to MEV in the ExPDB database as templates (consistency ≥ 30%) and then performs sequence alignment with the template sequence. The most crucial step is to replace the amino acids in the target sequence with their corresponding spatial positions in the template structure through sequence alignment and use homologous modeling software to predict the structural model. Finally, conduct a quality assessment, replace templates, and correct sequence alignment. Repeat this process until the model's quality is satisfactory [36].

**2.2.10 Molecular docking.** Molecular docking of MEV and TLR4 (PDB ID:4G8A) immunoreceptors using the HDOCK server (http://hdock.phys.hust.edu.cn/). Firstly, we submit the protein FASTA formats of TLR4 and MEV. The second step is sequence similarity search. Given a sequence from input or structural transformation, perform a sequence similarity search on the PDB sequence database to find homologous sequences of receptor and ligand molecules. During template selection, if the differences in sequence coverage, similarity, and

resolution between the two templates are within 10%, the template in the complex also takes precedence over the others. With the selected templates, the models are produced with the aid of MODELLER. Finally, traditional global docking, a layered docking program based on FFT, is used for assuming the binding direction of global sampling. Finally, we selected the advantage model from the top ten models [37]. Key interacting residues were inferred from LigPlot + generated protein-ligand 2D structural interaction maps and their 3D structures were visualized by PyMOL.

**2.2.11 Immunostimulation.** C-ImmSim (https://kraken.iac.rm.cnr.it/C-IMMSIM/) describes the different stages of the process of recognition and response of the immune system to pathogens, while the server simulates three different parts representing three separate anatomical regions in mammals, including the bone marrow, thymus and tertiary lymphoid organs (lymph nodes). The three different intervals are 1, 84, and 168 [38, 39]. In the Xinjiang population, HLA-A*1101, HLA-A*0201, HLA-B*5101, HLA-B*3501, HLA-DRB1*0701, and HLA-DRB1*1501 high-frequency alleles were selected for analysis [29]. Finally, the parameters were set to the default values of the software: the simulation parameter random seed was set to 12345, the simulation volume was set to 50 and the simulation step was set to 1050.

**2.2.12 Optimization of MEV codons and in silico cloning.** We used the online codon optimization tool ExpOptimizer (https://www.novopro.cn/tools/codon-optimization.html) to analyze and optimize the codons of MEV. For analysis and optimization, we selected E. coli as the expression host and excluded two restriction endonuclease sites, XHOI and BamHI. Then obtained the DNA sequence of the MEV codon was from the results. In silico cloning, we selected pET-28a (+) as the vector. Primers were then designed and polymerase chain reaction (PCR) was completed in SnapGene 6.1.1 with primer lengths between 15 and 30 bp, Tm values chosen at 60°C, an annealing temperature of 1°C and GC content between 40% and 60%. Finally, the appropriate nucleic acid endonuclease sites (XHOI and BamHI) in the polyclonal site (MCS) region were analyzed and the MEV-amplified target gene sequence was inserted into the vector to complete the in-silico cloning.

**2.2.13 agarose gel electrophoresis of MEV.** Agarose gel electrophoresis of the target gene (after PCR), vector, and recombinant plasmid was simulated in SnapGene 6.1.1and the experiments were completed in TBE buffer at a concentration of 1% agarose.

**2.2.14 Molecular dynamics simulation.** iMODS (http://imods.Chaconlab.org/) is an internal coordinate normal mode analysis (NMA) server. Using NMA to explore the aggregation motion of proteins and nucleic acids in internal coordinates (torsion space) and to explore the molecular dynamics of vaccines [40–42]. We performed molecular dynamics simulations of the MEV-TLR4 complex and analyzed the deformability and rigidity of each residue of the complex from the output results. Since the definition of the dihedral angle for backbone atoms N, CA, and C is mandatory, we submit the PDB file format of the MEV-TLR4 complex (including atomic coordinates). Next, we selected the CA model from three coarse-grained (CG) models. Because it can show that Cα atoms account for the whole residue mass. Then, we chose JAVA in the JSmol plugin, which is the fastest and most memory-effective mode. Finally, the other parameters remain unchanged. It is worth noting that Normal Mode Analysis (NMA) was selected [40].

# 3. Results

## 3.1 Target protein selection and sequence search

By bioinformatics analysis, the results of antigenicity analysis for VirB8 and VirB10 were 0.6221 and 0.6906, respectively, both greater than the threshold value of 0.4 with good antigenicity. The allergenicity result of VirB8 is PROBABLE ALLERGEN. The allergenicity result of

VirB10 is PROBABLE NON ALLERGEN. The combined (Table 1) results show that VirB8 and VirB10 are hydrophilic and stable proteins. The high precision of the MAFFT in Jalview (Fig 1) suggests that the two proteins share several homologies. Among them, the high homology sequences of VirB8 and VirB10 suggest that these proteins may be derived from the same gene and may play similar roles in the immune response.

## 3.2 Prediction of signal peptides

We applied SignalP5.0 and LiPOP1.0 to predict the signal peptide and the final result was no signal peptide for both VirB8 and VirB10 (Fig 2).

## 3.3 Prediction of T-cell epitopes

We selected the top 10 epitopes for each software score and applied VaxiJen to analyze the antigenicity of the epitopes; used AllergenFP v1.0 (http://www.ddg-pharmfac.net/AllergenFP/) to detect whether the epitopes were allergenic; and applied ToxinPred (https://webs.iiitd.edu.in/raghava/toxinpred/design.php) to predict the toxicity of the epitopes. Ultimately, two CTL-dominant epitopes and nine HTL-dominant epitopes were obtained, the remaining excluded epitopes were toxic or allergic (Tables 2 and 3). These epitopes were highly antigenic, non-toxic and PROBABLE NON ALLERGEN. The T cell analysis epitopes of VirB8 and VirB10 are reflected in the supplementary tables (S1–S8 Tables).

## 3.4 Prediction of B-cell epitopes

By bioinformatics analysis, we obtained six conformational B-cell epitopes and six dominant linear B-cell epitopes with a high score, non-toxic, and PROBABLE NON ALLERGEN (Tables 4 and 5). The B cell analysis epitopes of VirB8 and VirB10 are reflected in the supplementary table (S9 Table and S1 and S2 Figs). It is worth noting that the conformational epitope of B cells has no antigenicity or allergenicity (Fig 3).

## 3.5 Molecular docking of T-cell epitopes to HLA alleles

Assessing the structural association of HLA alleles with T cell epitopes. Results for CTL epitopes interacting with HLA-A*02:01, Docking Score: -242.47 Confidence Score: 0.8641 ligand RMSD (Å): 32.94. HTL epitopes interacting with HLA-DRB1*01:01 Results, Docking Score: -275.57 Confidence Score: 0.9249 ligand RMSD (Å): 150.52. The final indication is that the molecules have a good affinity for the docked complexes. (Fig 4).

## 3.6 The construction of MEV

Results of the MEV construction (Fig 5). The dominant epitopes selected for our MEV vaccine construction were 2 CTL epitopes, 9 HTL epitopes, 6 LBE epitopes, and 6 CBE epitopes.

**Table 1. Basic structure and physicochemical properties of amino acids.**

| amino acid | Number of amino acids | Molecular weight | Instability index | Grand average of hydropathicity (GRAVY) | Theoretical PI |
|---|---|---|---|---|---|
| VirB8 | 239 | 26445.83 | 28.56 | -0.392 | 8.60 |
| VirB10 | 380 | 40442.32 | 36.52 | -0.155 | 7.72 |

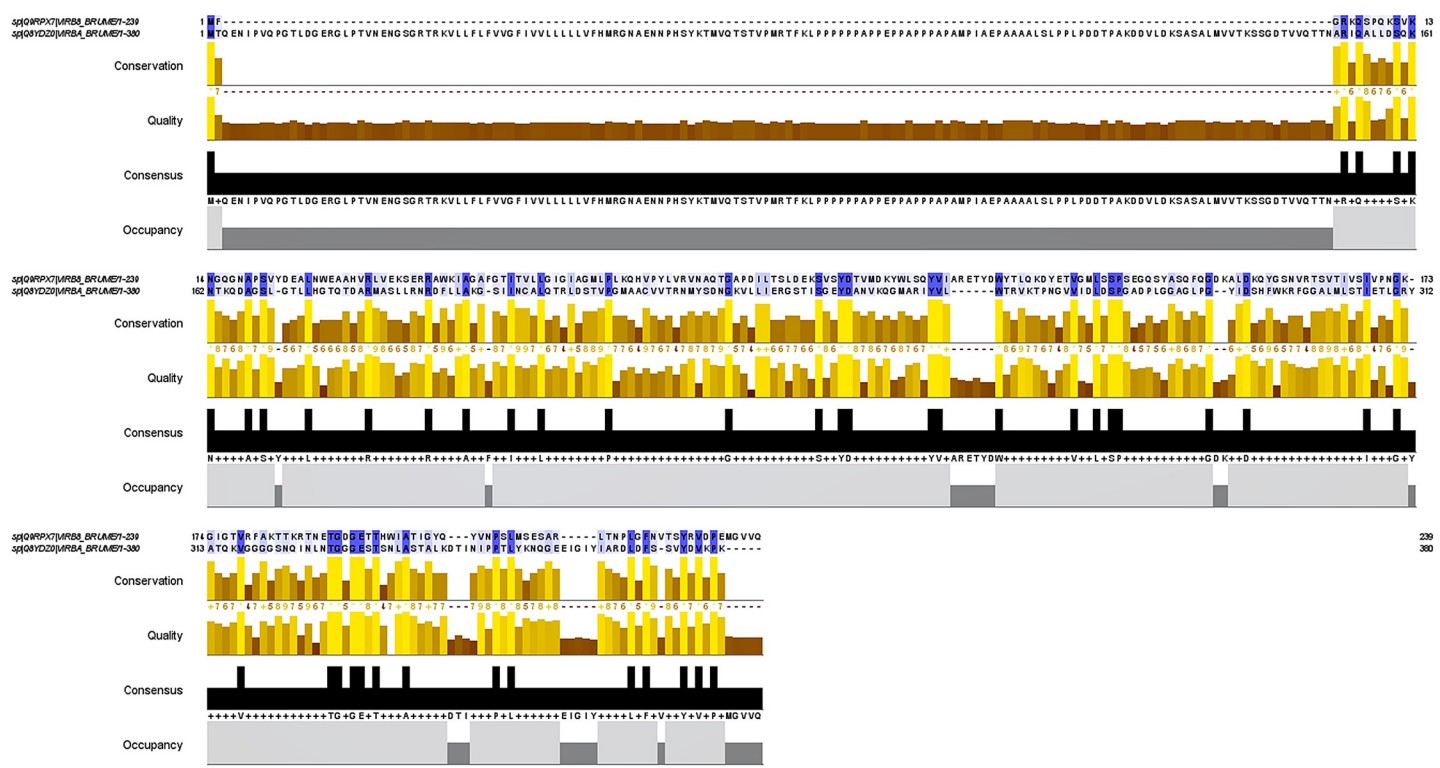

**Fig 1. Comparison of protein homologous sequences, the black regions are highly conserved amino acid regions, while the blue regions are similar amino acid sequence regions (darker colours represent high homology).**

## 3.7 Physicochemical properties, antigenicity, solubility and allergenicity of MEV

The molecular formula of MEV was $C_{2479}H_{3998}N_{720}O_{762}S_{14}$. The molecular weight of MEV was 56530.22KD. The number of amino acid residues in MEV was 537. MEV had a solubility of 0.99, which means that the protein antigen was soluble. The instability index (II) was computed to be 30.75, less than the threshold of 40, so MEV was a stable protein. The GRAVY was -0.626. Indicates that MEV was a hydrophilic protein. In addition, the antigenicity of MEV is 0.8788 (greater than the threshold value of 0.4), indicating that the protein is antigenic.

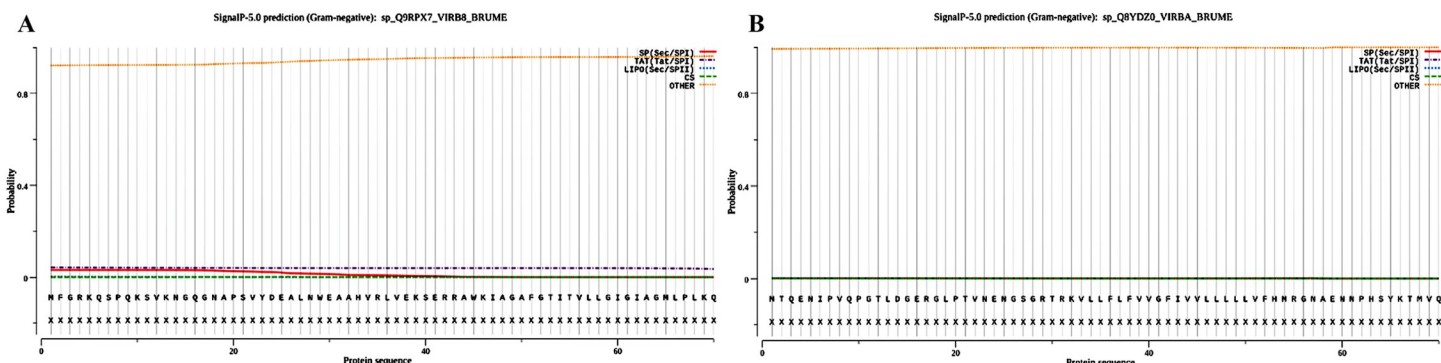

**Fig 2. Results of analysis using SignalP-5.0.SP (Sec/SPI)/LIPO (Sec/SPII)/TAT (Tat/SPI) (depending on what type of signal peptide is predicted); CS (the cleavage site) and OTHER (the probability that the sequence does not have any kind of signal peptide).**

**Table 2. Physical properties of CTL dominant epitopes and antigenicity.**

|  | CTL epitopes | Molecular weight | Instability index | Grand average of hydropathicity (GRAVY) | Theoretical PI | Antigenicity |
|---|---|---|---|---|---|---|
| VirB8 | SVSYDTVMDK | 1144.26 | 9.16 | -0.420 | 8.60 | 1.0037 |
| VirB10 | FKLPPPPPPA | 1060.30 | 120.30 | -0.510 | 7.72 | 1.8350 |

According to the allergenicity prediction results, PROBABLE NON-ALLERGEN. In conclusion, the MEV design is feasible.

### 3.8 Forecasts for secondary and tertiary structures

The predicted results show that in the secondary structure prediction, α-helix accounts for 28.12%, β-turn for 5.03%, random coil for 49.53%and extended strand for 17.32%, the ratio of the four is consistent with the tertiary structure (Fig 6A and 6B). Furthermore, we depicted the tertiary structure of MEV in Discover Studio, further demonstrating that the prediction of a tertiary structure is reasonable (Fig 6C). Ultimately, (Fig 6D) shows the regions where the donor and acceptor are likely to be present.

### 3.9 Quality assessment of models

A Ramachandran plot is a way to visualize energetically favoring regions for backbone dihedral angles against amino acid residues in protein structure. The number of observed Φ (Phi; C-N-CA-C) / Ψ (Psi; N-CA-C-N) pairs determines the contour lines (Fig 7). The dark green region in the Ramachandran plots indicated the allowed region (90.47%), the light green region in the diagram indicated the maximum allowed region (7.10%) and the blank region in the diagram indicated the disallowed region (2.43%). MolProbity Score:3.29, Clash Score:197.92. Overall, the quality of MEV's models was assessed as better.

### 3.10 Molecular docking

With the HDOCK server, We obtained molecular docking results for the top ten models. Comprehensive docking score and ligand RMSD, we selected the advantage model for the analysis of the interaction between receptor (TLR4) and ligand (MEV). The results showed a docking score of -298.12, a ligand RMSD of 102.33 Å, and a confidence score of 0.9508. The docking structure was demonstrated by Discovery Studio (Fig 8A). The 3D interaction structure was visualized by PyMOL (Fig 8B). The Ligplot predicts a two-dimensional interaction interface. There are 2 salt bridges with the red dotted line and 4 hydrogen bonds with the green dotted line in this two-dimensional interface (Fig 8C).

**Table 3. Physical properties of HTL dominant epitopes and antigenicity.**

|  | Serials | HTL epitopes | Molecular weight | Instability index | Grand average of hydropathicity (GRAVY) | Theoretical PI | Antigenicity |
|---|---|---|---|---|---|---|---|
| VirB8 | 226–240 | NVTSYRVDPEMGVVQ | 1693.89 | -6.03 | -0.307 | 4.37 | 1.1654 |
|  | 93–107 | DEKSVSYDTVMDKYW | 1866.03 | 2.53 | -1.153 | 4.23 | 0.8741 |
|  | 31–45 | EAAHVRLVEKSERRA | 1750.98 | 54.70 | -0.953 | 8.85 | 1.0387 |
|  | 32–46 | AAHVRLVEKSERRAW | 1808.08 | 54.70 | -0.780 | 10.74 | 0.5797 |
|  | 94–108 | EKSVSYDTVMDKYWL | 1864.10 | 10.75 | -0.667 | 4.56 | 0.6779 |
| VirB10 | 142–156 | SGDTVVQTTNARIQA | 1560.68 | -10.71 | -0.353 | 5.55 | 1.3987 |
|  | 143–157 | GDTVVQTTNARIQAL | 1586.77 | -10.71 | -0.047 | 5.84 | 1.0796 |
|  | 141–155 | SSGDTVVQTTNARIQ | 1576.68 | 2.13 | -0527 | 5.55 | 1.5812 |
|  | 266–270 | PNGVVIDLDSPGADP | 1465.58 | 30.13 | -0.127 | 3.42 | 0.8910 |

**Table 4. Physical properties of LBEs dominant epitopes and antigenicity.**

| | Serials | LBEs | Molecular weight | Instability index | Grand average of hydropathicity (GRAVY) | Theoretical PI | Antigenicity | Score |
|---|---|---|---|---|---|---|---|---|
| VirB8 | 18–37 | NAPSVYDEALNWEAAHVRLV | 2254.49 | 27.40 | -0.120 | 4.65 | 0.7330 | 0.86 |
| | 72–91 | VPYLVRVNAQTGAPDILTSL | 2127.47 | 26.13 | 0.500 | 5.81 | 0.6699 | 0.82 |
| VirB10 | 99–118 | PAMPIAEPAAAALSLPPLPD | 1942.30 | 85.48 | 0.560 | 3.67 | 0.4357 | 0.910 |
| | 329–348 | TGGGESTSNLASTALKDTIN | 1937.05 | 11.69 | -0.430 | 4.37 | 1.3876 | 0.890 |
| | 111–130 | LSLPPLPDDTPAKDDVLDKS | 2136.38 | 32.49 | -0.640 | 3.97 | 0.4408 | 0.890 |
| | 11–30 | GTLDGERGLPTVNENGSGRT | 2030.14 | -0.65 | -1.060 | 4.68 | 1.0657 | 0.890 |

## 3.11 Immunostimulation

The C-ImmSim server was used to simulate the immune response to three injections of the vaccine. In the secondary and tertiary immune responses, the concentrations of IgM and IgG continued to rise as the antigen decreased and the amount of IgM was consistently higher than IgG, peaking at the tertiary response (Fig 9A). B cells are mainly involved in humoral immunity and play an important role in the stimulated immune response, with the number of B cells increasing with the three doses of vaccine and eventually reaching a peak (Fig 9B). The growth trend of helper T cells after three doses of vaccine was similar to that of B cells, eventually reaching a peak (Fig 9C). Macrophage activity was enhanced with the three doses of vaccine (Fig 9G). In contrast, cytotoxic T cells (Fig 9D), natural killer cells (Fig 9E), dendritic cells (Fig 9F), and EP (Fig 9H) all showed relative stability in general. In addition, the vaccine injection induced a high response of cytokines and interleukins, resulting in significant elevations of

**Table 5. Physical properties of CBEs dominant epitopes and antigenicity.**

| | Serials | Residues | CBEs | Molecular weight | Instability index | Grand average of hydropathicity (GRAVY) | Theoretical PI | Score |
|---|---|---|---|---|---|---|---|---|
| VirB8 | 1–5 | M1, F2, G3, R4, K5 | MFGRK | 637.80 | 8.00 | -0.820 | 11.00 | 0.987 |
| | 235–239 | M235, G236, V237, V238, Q239 | MGVVQ | 532.66 | 8.00 | 1.280 | 5.28 | 0.868 |
| | 6–43 | Q6,S7,P8,Q9,K10,S11,V12, K13,N14,G15,Q16,G17, N18,A19,P20,S21,V22,Y23, D24,E25,A26,L27,N28,W29, E30,A31,A32,H33,V34,R35, L36,V37,E38,K39,S40,E41, R43 | QSPQKSVKNGQGNAPSVYDEALNWEAAHVRLVEKSER | 4123.51 | 46.43 | -1.086 | 6.77 | 0.839 |
| VirB10 | 139–147 | K139,S140,S141,G142,D143, T144,V145,V146,Q147 | KSSGDTVVQ | 919.99 | 13.59 | -0.578 | 5.84 | 0.949 |
| | 4–16 | E4,N5,I6,P7,V8,Q9,P10, G11,T12,L13,D14,G15,E16 | ENIPVQPGTLDGE | 1368.46 | 58.12 | -0.746 | 3.57 | 0.93 |
| | 91–134 | E91,P92,P93,A94,P95,P96, P97,A98,P99,A100,M101, P102,I103,A104,E105,P106, A107,A108,A109,A110, L111,S112,L113,P114,P115, L116,P117,D118,D119, T120,P121,A122,K123, D124,D125,V126,L127, D128,S130,A131,S132,A133, L134 | EPPAPPPAPAMPIAEPAAAALSLPPLPDDTPAKDDVLDKSASAL | 4326.93 | 78.36 | -0.118 | 3.85 | 0.822 |

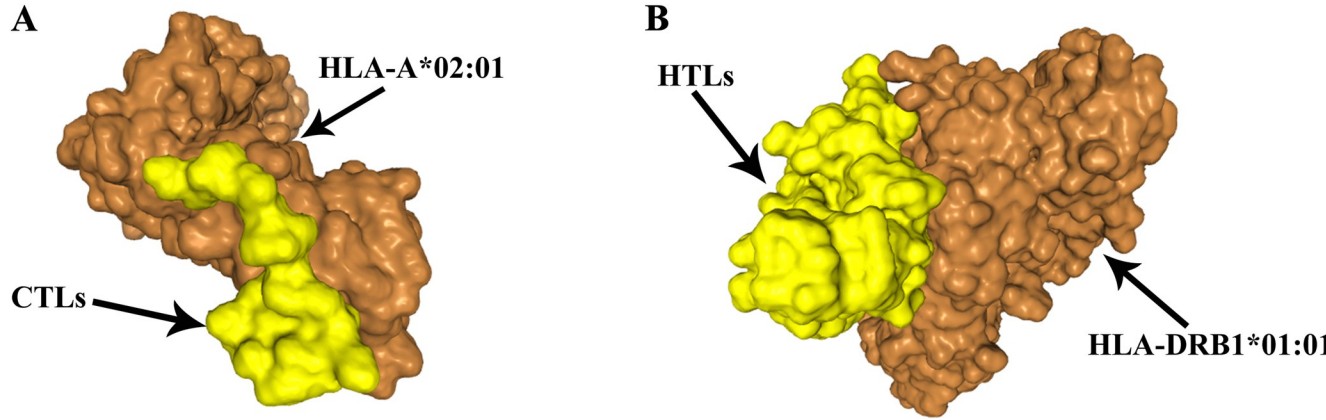

**Fig 3. B-cell conformational epitopes (A-C)** B-cell conformational epitope residues of VirB8. **(D-F)** B-cell conformational epitope residues of VirB10 **(A)** Residues: M1, F2, G3, R4, K5. **(B)** Residues: M235, G236, V237, V238, Q239. **(C)** Residues: Q6, S7, P8, Q9, K10, S11, V12, K13, N14, G15, Q16, G17, N18, A19, P20, S21, V22, Y23, D24, E25, A26, L27, N28, W29, E30, A31, A32, H33, V34, R35, L36, V37, E38, K39, S40, E41, R43. VirB10. **(D)** Residues: K139, S140, S141, G142, D143, T144, V145, V146, Q147. **(E)** Residues: E4, N5, I6, P7, V8, Q9, P10, G11, T12, L13, D14, G15, E16. **(F)** Residues: E91, P92, P93, A94, P95, P96, P97, A98, P99, A100, M101, P102, I103, A104, E105, P106, A107, A108, A109, A110, L111, S112, L113, P114, P115, L116, P117, D118, D119, T120, P121, A122, K123, D124, D125, V126, L127, D128, S130, A131, S132, A133, L134.

**Fig 4. The docked complexes. (A-B)** HLA-bacterial peptide complexes. **(A)** Results of molecular docking of CTL epitopes to HLA-A*02:01. **(B)** Results of molecular docking of HTL epitopes to HLA-DRB1*01:01.

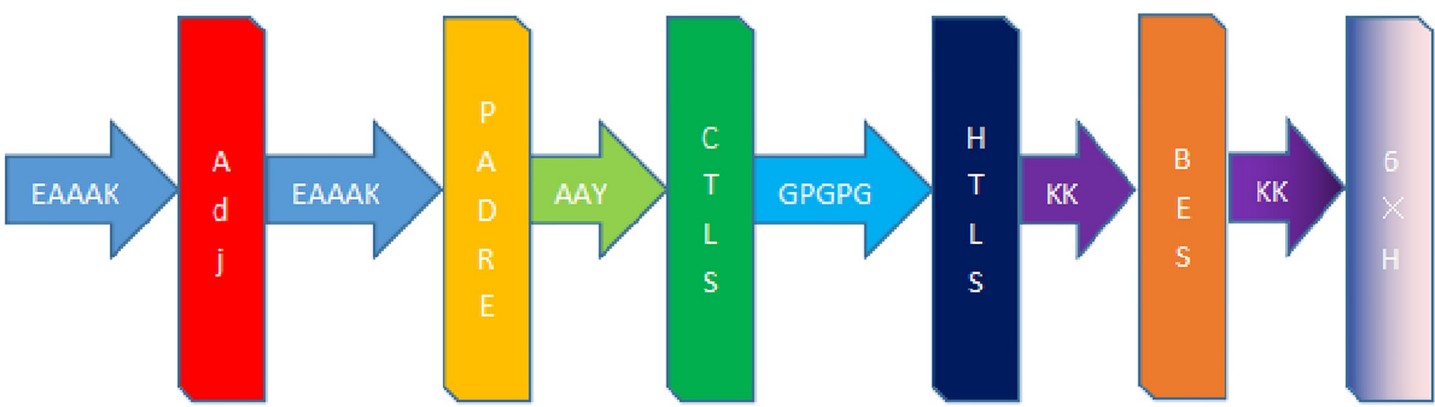

**Fig 5. Amino acid sequence of MEV.** EAAK, AAY, GPPGPG and KK are linkers."Adj" in red is β-defensin-3. The Yellow "PADRE" is the PADRE sequence. Dark green"CTLS" represents the dominant epitope of the selected cytotoxic T cells. The dark blue "HTLS" represents the dominant epitope of the selected helper T cells. "BES" represents the dominant epitope of the selected linear and conformational B cells. The "6×H" indicates a polyhistidine tag.

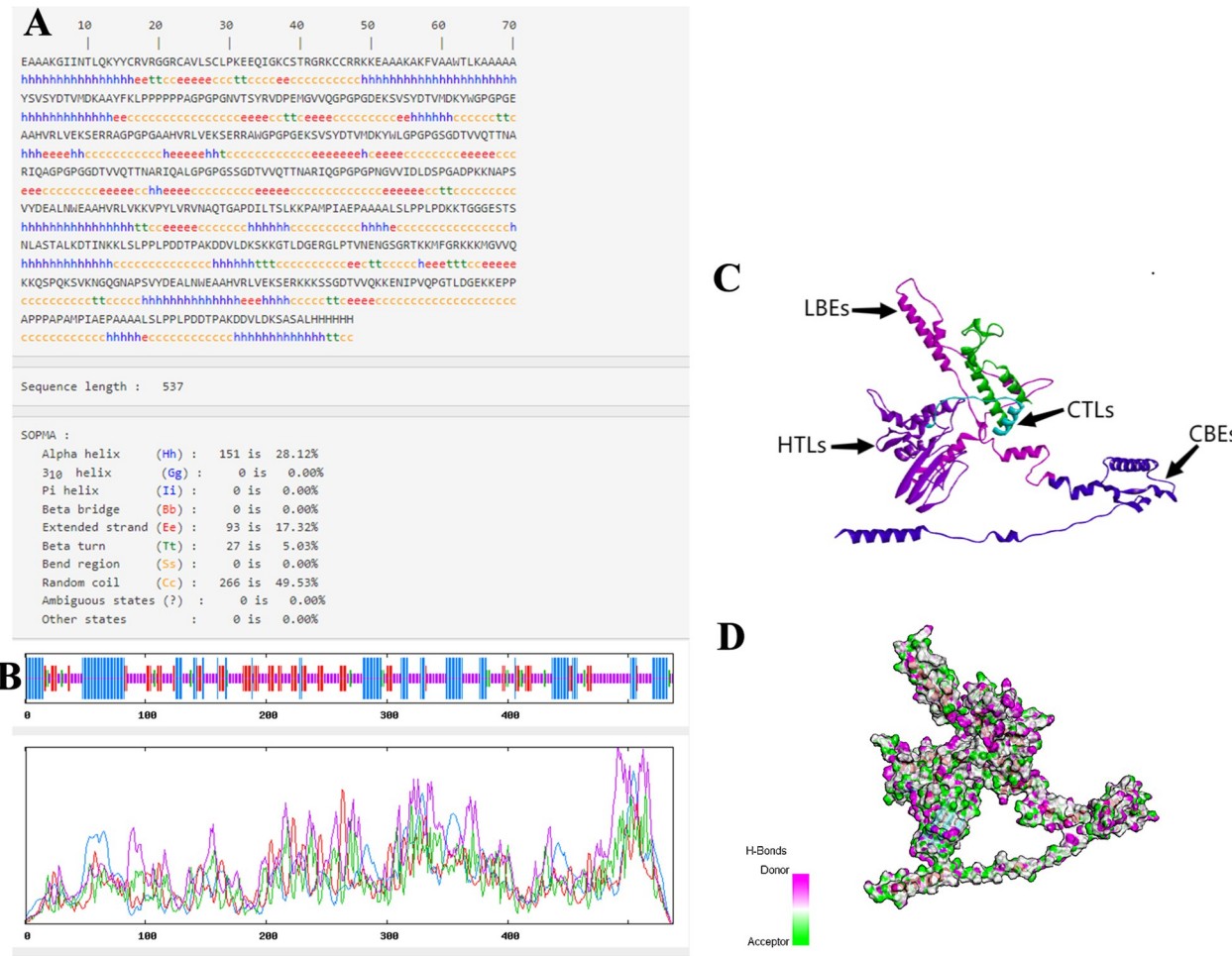

**Fig 6. The predicted results of MEV. (A-B)** Predicted results for MEV secondary structure. **(C)** Predicted results for the MEV tertiary structure. **(D)** H-Bonds of MEV. As illustrated in the figure, the "pink area" stands for the donor, and the "green area" stands for the acceptor.

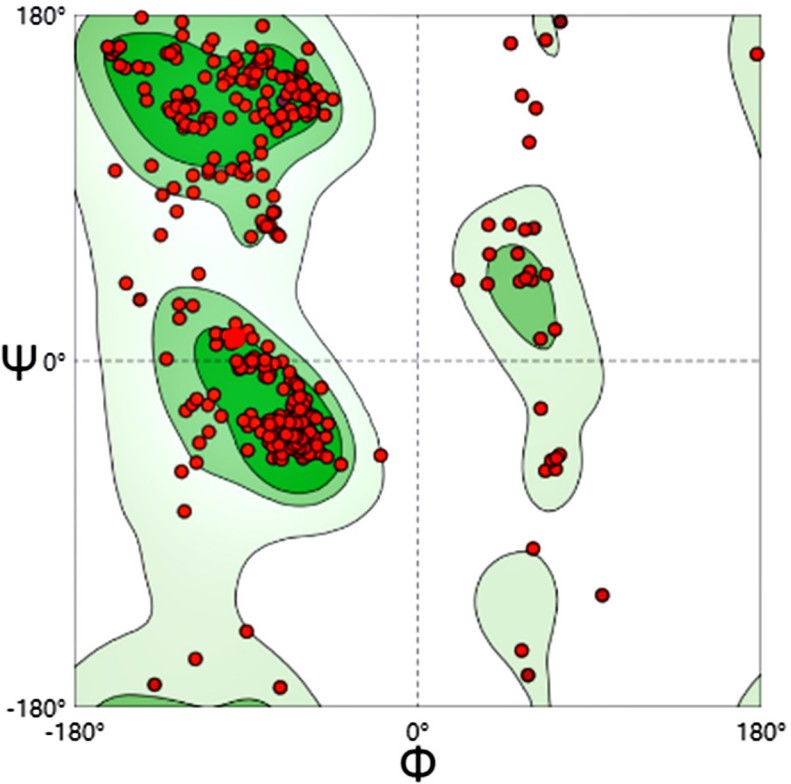

**Fig 7. Validation: Ramachandran plot analysis showing 90.47% in favored, 7.10% in allowed, and 2.43% in disallowed regions of protein residues.**

IFN-γ, TGF-β, IL-10, IL-12, and IL-2. Finally, the danger signal was extremely low, indicating a difference in immune response (Fig 9I).

## 3.12 Optimization of MEV codons and in silico cloning

The quality of the codon optimization was measured by the display of the codon adaptation index (CAI) and GC content. The optimized CAI value for MEV was 0.80. The ideal range for GC content was 30%-70% and the optimized GC content for MEV was 58.35%, which was within the ideal range (Fig 10A and 10B). Based on the principles of primer design, a forward primer (5′–CTCGAGGAAGCGGCGGC–3′) with a length of 17, a Tm value of 62and a GC content of 76% and a reverse primer (5′–GGATCCATGGTGGTGATGATGGTGC–3′) with a length of 25, a Tm value of 63and a GC content of 56% were designed. The target gene for MEV was then amplified in SnapGene (Fig 10C). Restriction endonuclease sites XHOI and BamHI were inserted into the N and C ends of the optimized codons at the time the primers were designed. Finally, these codons are inserted into the MCS structural domain in the vector and should be considered to correspond to the restriction endonuclease sites (XHOI and BamHI) on the vector (Fig 10D).

## 3.13 Agarose gel electrophoresis of MEV

As shown in the figure (Fig 11). The amount of each DNA was consistent with previous predictions, with 1611 bp of MEV sequence,1623 bp of MEV sequence after PCR, 5639 bp of pET-28a (+) sequence, and 6946 bp of recombinant plasmid sequence.

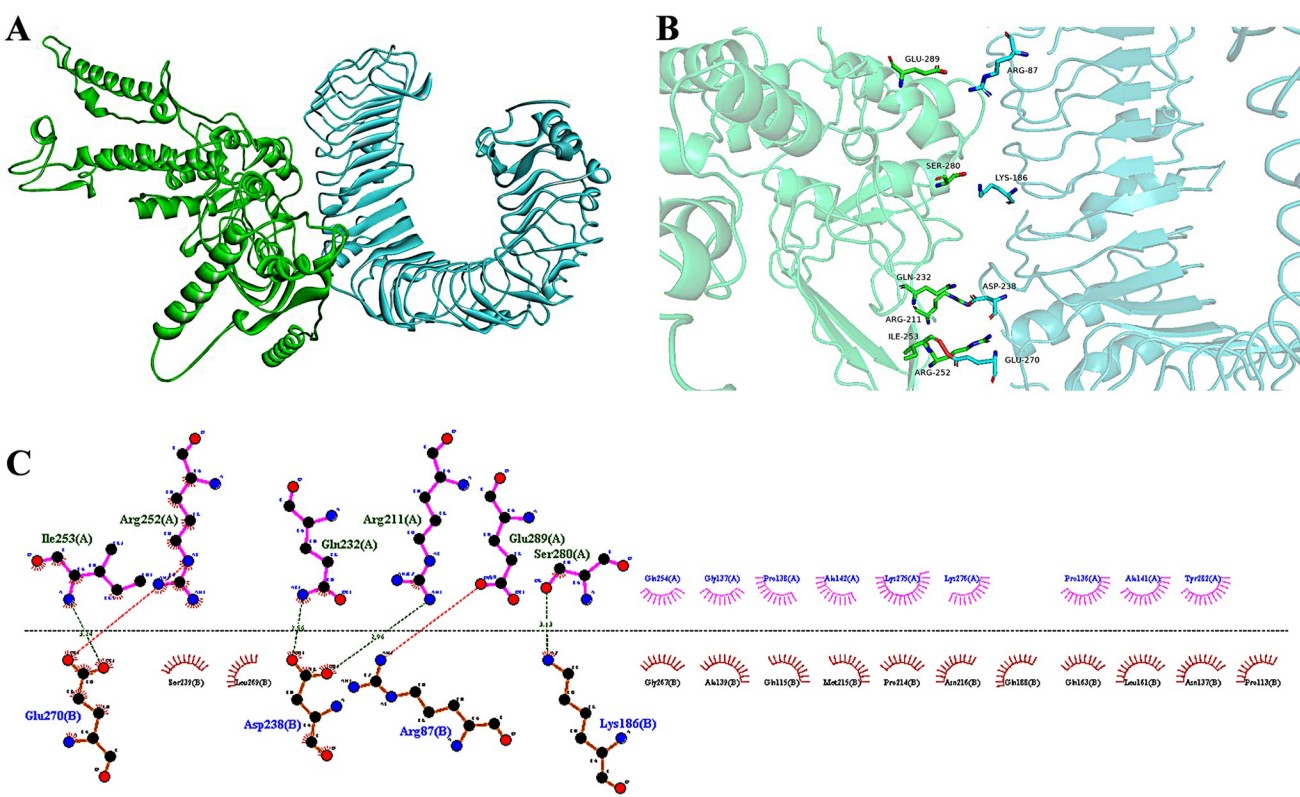

**Fig 8. The results of molecular docking. (A)** Structural presentation of the MEV-TLR4 complex using Discovery Studio: MEV donor in green and TLR4 acceptor in blue. **(B)** Analysis of the interaction of the MEV-TLR4 complex and its 3D image taken using PyMol. **(C)** Analysis of the interaction of the MEV-TLR4 complex and its 2D image taken using Ligplot.

### 3.14 Molecular dynamics simulation

The molecular motion was analyzed by normal modal analysis by iMODs and the results are shown in Fig 12. Additionally, using counting normal mode analysis (NMA) via server iMODS, the stiffness of the mobility and deformability of the complex residues were examined. Less mobility was shown in the complex's deformability plot for the residues and the complex's B-factor was less deviated by the NMA under analysis (Fig 12A and 12C). Individual (purple) and cumulative (green) variations are depicted as colored bars. The individual variance of each successive mode exhibited a modest decline, according to the variance plot analysis (Fig 12B). 9.353365e-05 was the calculated eigenvalue for the complicated structure and it grew steadily in each mode [42] (Fig 12D). The covariance matrix of the complex molecule represents the degree of correlation between different pairs of atomic motions, which can be either correlated (red), anti-correlated (blue), or uncorrelated (white) (Fig 12E). Additionally, an elastic network model of MEV was generated to differentiate the atom pairs connected by springs. In the diagram, each dot represents a spring between the corresponding atom pairs, and, the color of the dot indicates the stiffness of the spring. The darker gray color indicates that the springs are stiffer (Fig 12F). These results indicate that the MEV-TLR4 complex has good stability and stiffness [43].

At present, there is still no good vaccine available for people suffering from brucellosis. Therefore, more and more people are designing vaccines targeting the outer membrane proteins of Brucella, such as Omp22, Omp19, Omp28, Omp10, Omp25, Omp31and BtpB. However, most vaccine designs are aimed at destroying the Brucella cell membrane and thereby

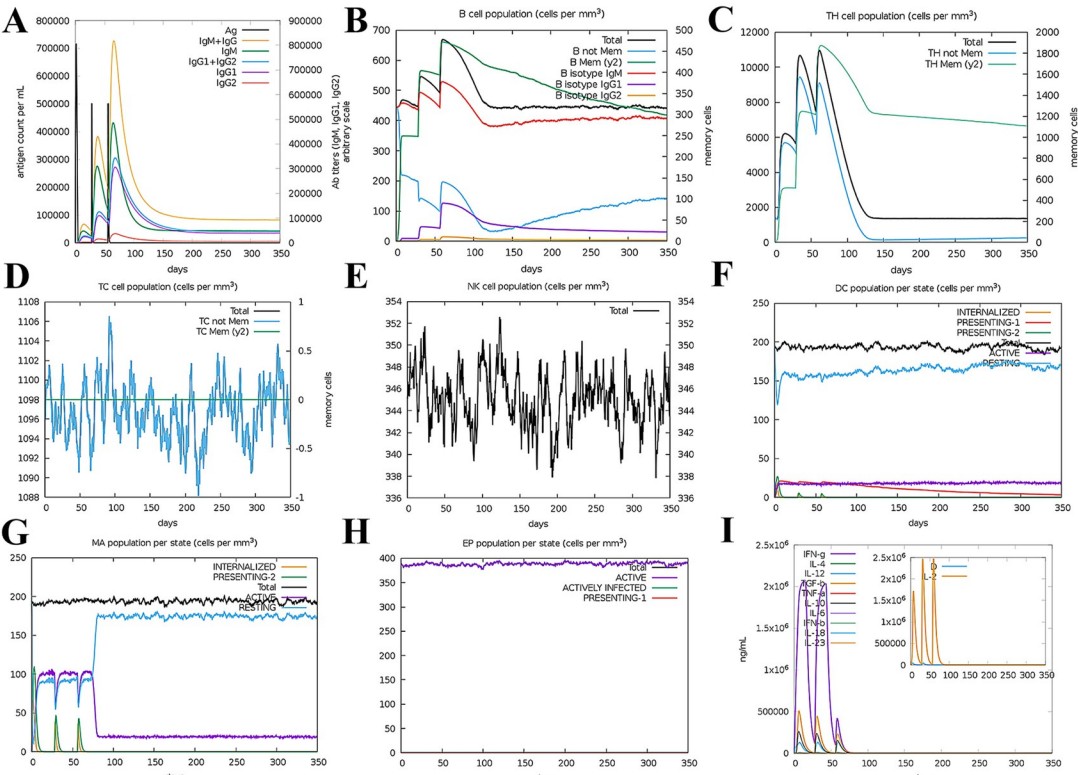

**Fig 9. The results of C-ImmSim. (A)** The immunoglobulin production after antigen injection. **(B)** The B cell population after three injections. **(C)**The Helper T Cell Population after three injections. **(D)** The Cytotoxic T Cell Population after three injections. **(E)** The NK-cell Population after three injections. **(F)**The Dendritic cell Population per state after three injections. **(G)** Macrophage Population per state after three injections. **(H)** The EP Population per state after three injections. **(I)** Concentration of cytokines and interleukins. Inset plot shows danger signal together with leukocyte growth factor IL-2.

killing bacteria [44–46]. It is worth noting that this study designed a vaccine for Brucella T4SS in order to disrupt the basic structure of T4SS and cause the bacteria to lose the conditions for intracellular reproduction. Therefore, it blocks the further infection of bacteria in the patient's body, adding a new protective pathway to protect the patient from the persistent infection of Brucella [47].

## 4. Discussion

Brucellosis is a debilitating zoonotic disease that can cause significant economic losses in livestock populations worldwide [48, 49]. As there is currently no Brucella vaccine for humans. Therefore, it is vital to develop a more effective and safer vaccine for humans [50]. We have constructed a new specific multi-epitope vaccine based on the structural basis of the Brucella type IV secretion system. The type IV secretion system (T4SS), encoded by the VirB manipulator, is an important virulence factor for Brucella, while its core component consists of VirB6--VirB10 [17]. Obviously, VirB8 is a two-site endosomal protein that plays a critical role in the nucleation of T4SS channels [51, 52]. Similarly, VirB10 is a bilayer protein inserted into the bacterial endosome and the proline-rich region plays a key role in core complex assembly and substrate secretion [53, 54]. Following previous studies, the proteins required to construct novel MEVs must be highly antigenic. We found that VirB8 and VirB10 are good choices for constructing multi-epitope vaccines because they both have high Antigenicity [55]. In

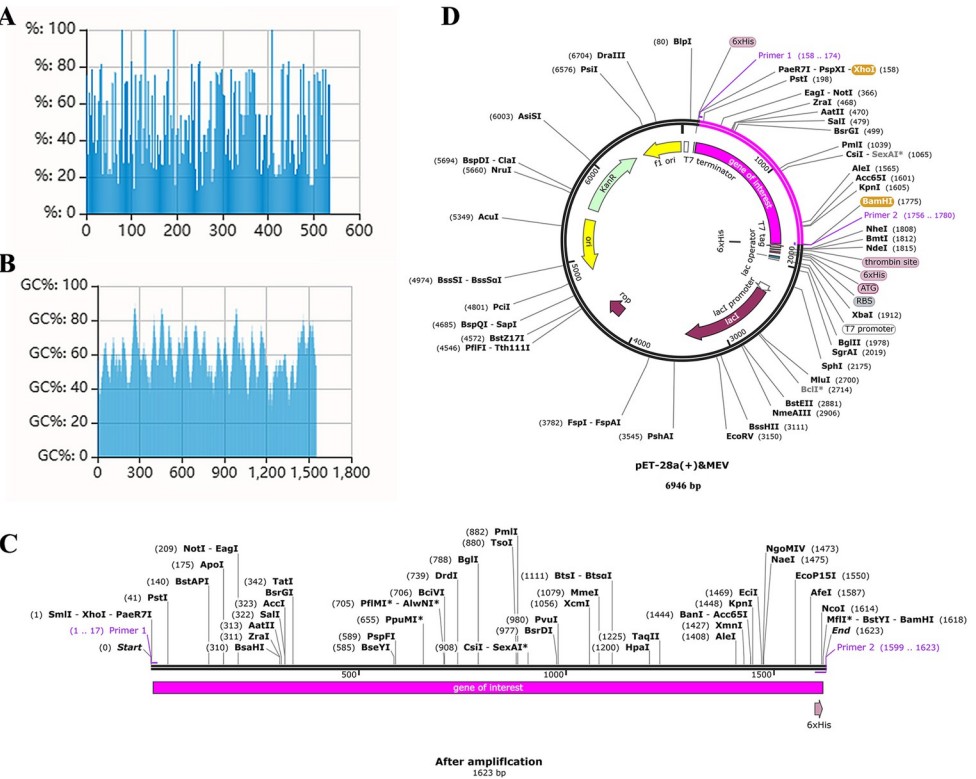

**Fig 10. Codon optimization of MEV and construction of plasmid vectors.** (A) CAI after codon optimization: 0.80. (B) GC content after codon optimization: 58.35%. (C) MEV after polymerase chain reaction. (D) The pink sequence (gene of interest) is the MEV codon sequence optimized after insertion into the vector (pET28a (+)). The cloning was done in SnapGene6.1.1.

addition, we analyzed their physical and chemical properties, including antigenicity, stability, and sensitization. Surprisingly, these results all indicate their potential for constructing multi-epitope vaccines. Ultimately, VirB8 and VirB10, the core components of the type IV secretion system (T4SS), were selected to construct a novel multi-epitope vaccine. On the other hand, the homology of the two proteins was verified during sequence alignment and met the requirements for novel vaccine design. Remarkably, there are no reports of MEV construction based on two proteins, VirB8 and VirB10.

In our study, signal peptides were predicted for VirB8 and VirB10 using SignalP5.0 and LiPOP1.0. Signal peptides usually contain 15–30 amino acids [56, 57]. They are usually located at the N-terminal end of the protein and influence the start of protein translation and the different primary structures of the signal peptide even influence protein folding and translocation [57, 58]. The expression level of a protein can be altered by replacing the signal peptide [57, 59]. Surprisingly, neither of these two proteins has a signal peptide and we do not need to deliberately remove the signal peptide sequence, which further proves that the target protein we selected is correct. Therefore, we believe that we can proceed with the next step of analysis.

In addition, the main goal to be achieved with vaccines is to provide lasting memories. It is therefore crucial to activate B cells and T cells to achieve this aspect. Meanwhile, helper T lymphocytes initiate humoral and cell-mediated immune responses; cytotoxic T lymphocytes prevent virus transmission by killing virus-infected cells and producing antiviral cytokines;and B lymphocytes are primarily involved in humoral immune responses [60, 61]. To predict the

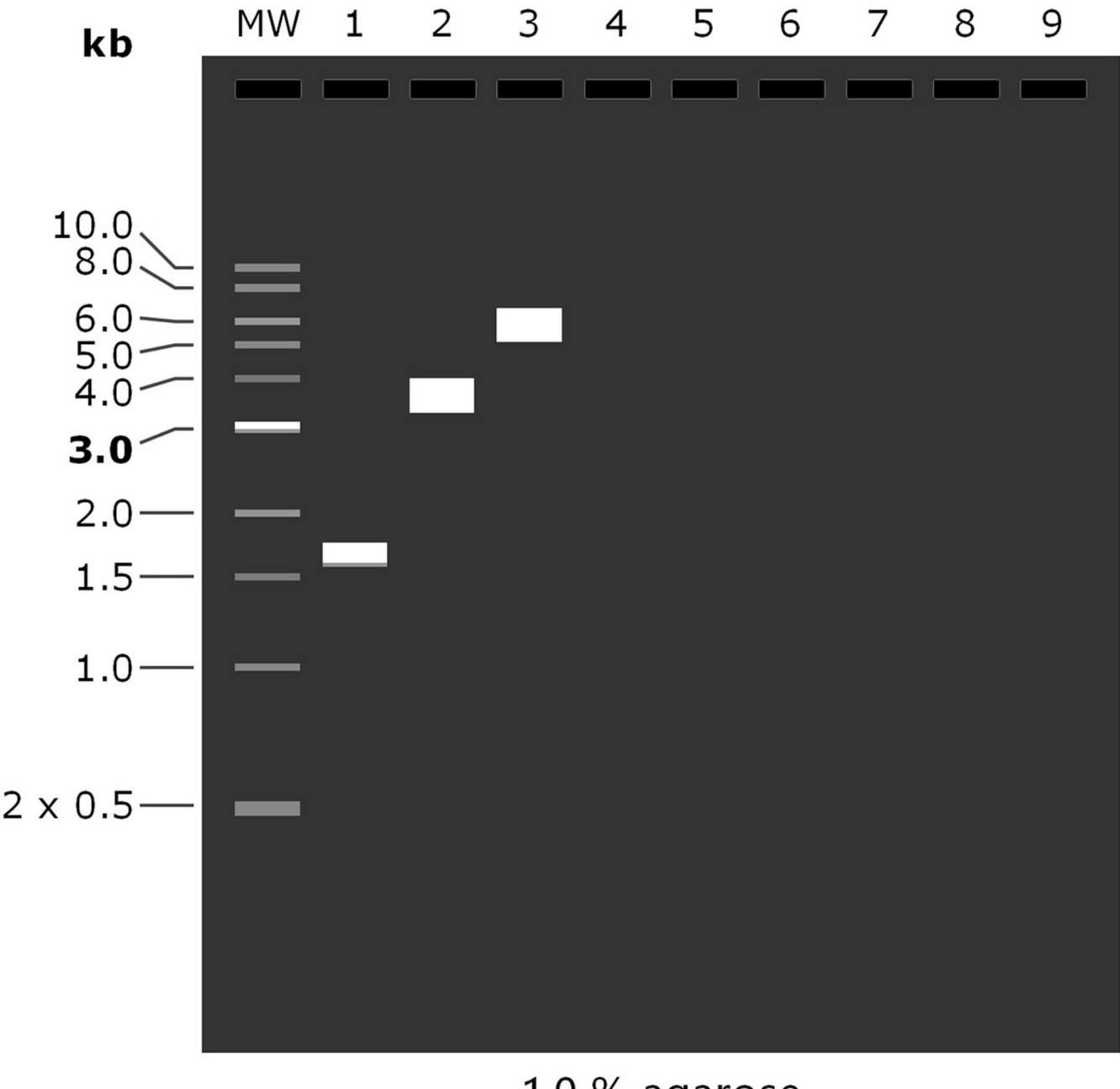

**Fig 11. Mock agarose gel electrophoresis results.** "1" represents MEV-PCR, "2" represents pET-28a (+), "3" represents pET-28a (+) &MEV recombinant plasmid.

epitopes of CTL, HTL, LBEand CBE and to select suitable candidate vaccines, different databases, and online servers were used [62, 63].

A multi-epitope vaccine consisting of CTL, HTLand B-cell epitopes triggers broad immune protection, and initiates cellular and humoral immune responses [64–66]. Then, we screened out 2 dominant CTL epitopes from both proteins using IEDB and NetCTLpan1.1, 9 dominant

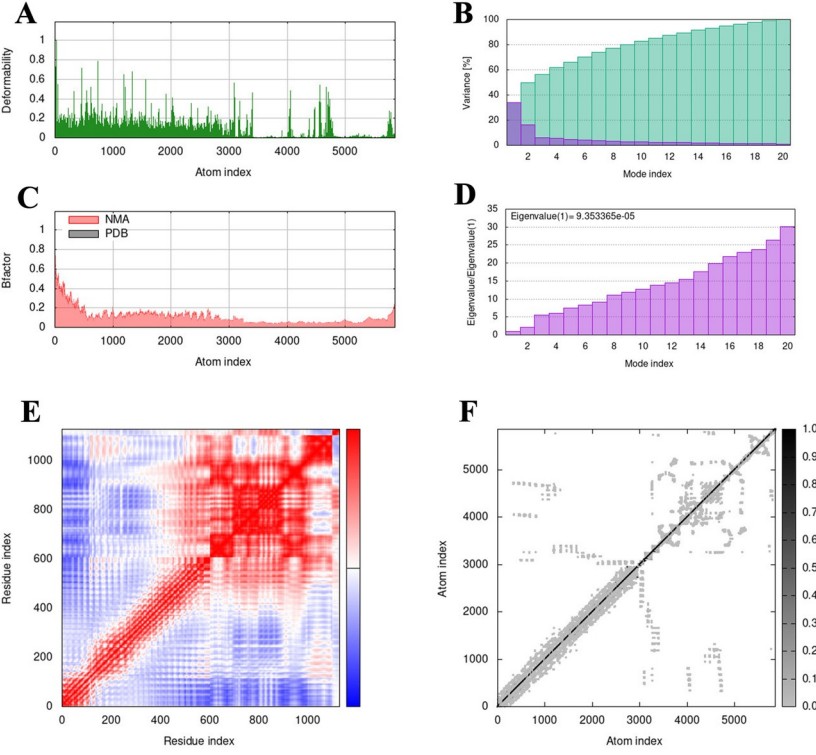

**Fig 12. The results of IMODs. (A-F)** Molecular dynamics simulation results. **(A)** Deformability values. **(B)** The variance associated to the modes. **(C)** B-factor and NMA graph. **(D)** Eigenvalues plot. **(E)** Covariance matrix graph. **(F)** The elastic network models.

HTL epitopes using IEDB and NetMHC-IIpan-4.0, 6 B-cell linear epitopes using ABCpredand 6 B-cell conformational epitopes using Ellipro of IEDB. Our MEV was then constructed by selecting the superior epitopes obtained above. In vaccine construction, the dominant epitopes are connected by linkers. We linked the CTL, B-cell, and HTL epitopes to the AAY, KK, and GPGPG linkers, respectively. The linker ensures that each epitope can trigger the immune response independently and avoids the creation of new epitopes that interfere with the immune response induced by the original epitope [67]. However, the immunogenicity of multi-epitope vaccines is poor when used alone and requires adjuvants for coupling [68]. Adjuvants are important components of vaccine formulations, preventing infection, influencing the specific immune response to antigens, maintaining the stability of peptides, and enhancing their immunogenicity [69]. A cationic peptide called human-defensin 3 (hBD3) has immunomodulatory effects on both innate and acquired immune responses. Furthermore, hBD3 can enhance the immunogenicity of epitope vaccines [70]. Thus, The adjuvant human beta-defensin-3 (hBD3) was fused to the N terminus with the help of the EAAAK linker. In addition, the PADRE peptide can not only induce CD4 T cells but also enhance the immune function of vaccine constructs. Meanwhile, the PADRE sequence can reduce human HLA-DR polymorphism [33]. Finally, the histidine sequence was added to obtain the complete MEV. Molecular docking between the HLA allele and the T cell epitope demonstrates the good affinity of the docking complex.

In structure-based reverse vaccinology, the protein molecular weight of our designed vaccine is 56 KD, which is in the ideal range (<110 KD) [71]. The theoretical pI of the vaccine

construct was 9.39 and the number of amino acids was 537, indicating the basic nature of the vaccine construct. Instability index and GRAVY values indicate vaccine protein stability and hydrophobicity. Additionally, an assessment of allergenicity and antigenicity showed that the vaccine was and high antigenicity (antigenicity of 0.8788) and that it was not allergenic. These results show that our vaccine constructs are stable, hydrophilic, antigenic, soluble, and non-sensitizing. In the next secondary structure predictions, β-turns and random coils account for 5.03% and 49.53%, respectively. The high proportion of beta-turned and random coils in MEV suggests that the vaccine protein may form antigenic epitopes [46]. The tertiary structure of the MEV was predicted by the RoseTTAFold server and the quality of the tertiary structure of the MEV was verified by the SWISS-MODEL structural assessment service. The results show that the three-stage structure of the MEV has a high degree of accuracy and a high approximation factor and that the overall structure is reliable and of good quality. Noticeably, the predicted β-turn angles and random coils are consistent with the secondary structure predictions, which further suggests that our vaccine constructs are correct. Strong interaction between antigenic molecules (MEV) and immune receptor molecules (TLR4) is necessary to initiate an immune response [72, 73]. Toll-like receptor 4 (TLR4), an innate immune receptor, is commonly involved in multi-epitope vaccine construction [74]. Then, protein-ligand docking analysis was performed on the MEV-TLR4 complex to examine the stability between the protein and TLR-4. In the atomic interaction diagram it is shown that there are strong interactions between molecules so that they can be transported throughout the host body [75].

Afterward, we used iMODS simulations to explore the stability of the complexes, generating eigenvalue data showing the stiffness and energy required to move the docked complexes. To verify that this vaccine structure can be involved in the humoral and cellular responses studied, we performed immune simulations of vaccine effects [76]. With the injection of three doses of vaccine, we found that T and B cells in the body increased with the number of injections and peaked at the third vaccination. Furthermore, MEV increased the levels of cytokines (IFN-γ, TGF-β, IL-10, IL-12and IL-2), IgG and IgM. Surprisingly, cytokine INF-γ The most obvious increase was in. IFN-γ indicates cell-mediated immunity, a chemokine that supports B cell proliferation, Ig isotype switching and humoral responses. Antigen-presenting cells display HTL epitopes when using MHC class II molecules and the HTL epitopes produce associated cytokines (IFN-γ, IL-10) to kill pathogens until they are completely eliminated. Therefore, IFN-γ's significant increase in the vaccine level indicates that the vaccine can play an important role in killing pathogens [77–79]. These results suggest that MEV can be designed to trigger a robust immune response without producing an allergic reaction and could be considered an excellent candidate for a brucellosis vaccine.

Efficient expression of the MEV vaccine protein in the Escherichia coli system is essential for the production of recombinant proteins [80]. We use the online codon optimization tool ExpOptimizer to optimize the amino acid sequence of the vaccine for various parameters such as 5' region optimization (translation initiation efficiency), DNA repeat optimization, and GC content optimization [81, 82]. The optimized codon GC content (58.35%) and codon adaptation index (CAI = 0.80) showed a good probability of vaccine protein expression levels in E. coli hosts. XhoI and BamHI restriction sites were then added to the 5' and 3' ends of the codon sequence and primers were designed for them to facilitate the polymerase chain reaction of the target gene. The final vaccine sequence was then cloned into the pET28a (+) vector, yielding a 6946-bp recombinant plasmid. Eventually, mock agarose gel electrophoresis experiments were performed on the target gene, vector, and recombinant plasmid.

Overall, MEV exhibits desirable physicochemical properties and an immune response. Molecular dynamics simulations demonstrate the high stability of MEV. Immunosimulations show that MEV triggers an immune response consistent with our hypothesis. In summary,

this study employs a wide range of bioinformatics methods to fight against Brucella infection, which offers a theoretical foundation for future lab tests.

## Supporting information

**S1 Table. MHC-I binding prediction results of VirB8 (IEDB).**
(DOCX)

**S2 Table. MHC-I binding prediction results of VirB8 (NetCTLpan version 1.1).**
(DOCX)

**S3 Table. MHC-I binding prediction results of VirB10 (IEDB).**
(DOCX)

**S4 Table. MHC-I binding prediction results of VirB10 (NetCTLpan version 1.1).**
(DOCX)

**S5 Table. MHC-II binding prediction results of VirB8 (IEDB).**
(DOCX)

**S6 Table. MHC-II binding prediction results of VirB8 (NetMHCIIpan version 4.0).**
(DOCX)

**S7 Table. MHC-II binding prediction results of VirB10 (IEDB).**
(DOCX)

**S8 Table. MHC-II binding prediction results of VirB10 (NetMHCIIpan version 4.0).**
(DOCX)

**S9 Table. LBEs results of VirB8 andVirB10 (ABCpred).**
(DOCX)

**S1 Fig. CBEs results of VirB8 (IEDB).**
(TIF)

**S2 Fig. CBEs results of VirB10 (IEDB).**
(TIF)

**S1 File. The minimal anonymized data.**
(ZIP)

## Acknowledgments

The authors are thankful to the State Key Laboratory of Pathogenesis, Prevention, Treatment of Central Asian High Incidence Diseases, The First Affiliated Hospital of Xinjiang Medical University, PR China.

## Author Contributions

**Data curation:** Zhengwei Yin.

**Software:** Zhengwei Yin, Min Li, Ce Niu, Mingkai Yu, Xinru Xie, Gulishati Haimiti, Wenhong Guo, Juan Shi, Yueyue He.

**Visualization:** Zhengwei Yin, Min Li, Ce Niu, Mingkai Yu, Xinru Xie, Gulishati Haimiti, Wenhong Guo, Juan Shi, Yueyue He.

**Writing – original draft:** Zhengwei Yin.

**Writing – review & editing:** Jianbing Ding, Fengbo Zhang.

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
