## [Decision Letter · Decision Letter 0]

29 May 2023

PONE-D-23-13551Design of multi-epitope vaccine candidate against Brucella type Ⅳ secretion system（T4SS）PLOS ONE

Dear Dr. Zhang,

Thank you for submitting your manuscript to PLOS ONE. After careful consideration, we feel that it has merit but does not fully meet PLOS ONE’s publication criteria as it currently stands. Therefore, we invite you to submit a revised version of the manuscript that addresses the points raised during the review process.

We look forward to receiving your revised manuscript.

Kind regards,

Sheikh Arslan Sehgal, PhD

Academic Editor

PLOS ONE

Journal Requirements:

https://www.nature.com/articles/s41598-022-14427-z?code=c1a18d7d-fe33-44c7-afd3-cfc6b510723d&error=cookies_not_supported

https://www.mdpi.com/2076-393X/9/10/1079/html

https://www.mdpi.com/1648-9144/58/10/1356/review_report

In your revision ensure you cite all your sources (including your own works), and quote or rephrase any duplicated text outside the methods section. Further consideration is dependent on these concerns being addressed.

   "This study was supported by grants (No. 81860352, No.81860375, No.81560322) from the National Natural Science Foundation of China and funds for the Xinjiang Key construction Project of the 13th Five-Year Plan (basic medicine) and Tianshan Youth Talent Program of Xinjiang Uygur Autonomous Region."

Reviewers' comments:

Reviewer's Responses to Questions

**Comments to the Author**

1. Is the manuscript technically sound, and do the data support the conclusions?

Reviewer #1: Yes

Reviewer #2: No

2. Has the statistical analysis been performed appropriately and rigorously? 

Reviewer #1: Yes

Reviewer #2: N/A

3. Have the authors made all data underlying the findings in their manuscript fully available?

Reviewer #1: Yes

Reviewer #2: No

4. Is the manuscript presented in an intelligible fashion and written in standard English?

Reviewer #1: Yes

Reviewer #2: No

5. Review Comments to the Author

Reviewer #1: The authors have done tremendous work entitled "Design of multiepitope vaccine candidate against Brucella type Ⅳ secretion system（T4SS)" in the field of vaccine development. There is space to improve the manuscript to be published in this prestigious journal. Here are some suggestions which will improve the quality of the manuscript.

Comments

1. The abstract should start with the introduction of the bacteria that the authors selected. For a clear understanding, add two introduction lines to the bacteria in the abstract.

2. The overall abstract is in the form of methodology. What promising results about the vaccine or epitopes the authors get are lacking in the abstract. The abstract should be modified according to the results and a clear conclusion.

3. The aim of the study should be discussed, unlike the methodology in the introduction. For better understanding, provide a clear study goal of why the vaccine is essential for the selected bacteria.

4. In the "1.1 Material sources" section, the authors retrieve the sequences from the UniProt database, but no ID is mentioned of any of the sequences, and also provide the UniProt database link.

5. In the section "1.2.1 Selection of target proteins", the authors find the antigenicity only for the selected proteins. First, discuss which proteins are selected from the pathogen and calculate the allergenicity of the proteins.

6. The sequence search and material sources sections should be combined in one subtopic and presented at the start of the methodology.

7. In the "Prediction of protein T-cell epitopes" section, the authors took six alleles for CTL prediction and three alleles for HTL prediction. Provide the population coverage analysis of these alleles to justify the selection of these alleles.

8. Explain how the final epitopes were selected from the prediction in the "Prediction of protein T-cell epitopes" section.

9. Similarly, in the B-cell predictions elaborates the process of final epitopes selection from the prediction.

10. The selected epitopes' physiochemical properties, antigenicity and allergenicity, should be calculated and reported as a safe selection for the Multiepitope vaccines.

11. The authors used the MEV-TLR4 complex in section "1.2.15 Molecular dynamics simulation", but did not mention any docking protocols or details of the Multiepitope vaccine and TLR4 receptor.

12. The section "1.2.15 Molecular dynamics simulation" is very poorly explained. The simulation protocols used by the server should be explained in detail, like which forcefields are used and what other parameters are provided to the simulation.

13. The authors stated in lines 210-211 that "the deformability, stiffness and stability of the complex with the output results". What analysis they performed after the simulation for what purpose should be adequately explained, like RMSD, RMSF, Radius of gyration, hydrogen bond analysis, PCA, etc.

14. The figure captions should be with the figure to explain it adequately, not in the manuscript's text.

15. The section "2.10 Quality assessment of models" should be appropriately discussed. Provide all the details of how much the favoured region is, allowed and disallowed regions of the model in the text.

16. The section "2.11 Molecular docking" is very poorly explained. Explain how many hydrogen bonds are formed between the MEV and TLR4 and how many salt bridges are formed.

17. The authors select the monomer of the TLR4 receptor, but its active form is the dimmer. Is there any specific reason for the monomer selection? Explain it, please, in the methodology.

18. The authors stated in lines 397-399 that "The eigenvalue in the eigenvalue plot is 9.353365e-05, which, according to previous studies, indicates that the complex has low deformability and good stability" but did not provide any sufficient references to the sentence.

19. The selection "2.15 Molecular dynamics simulation" results are discussed very poorly. Kindly search for some literature on how to explain these results in a better way. The output of the results is explained like a methodology. Focus on what good results you obtained from the output and discuss those just.

20. Add vaccine development studies for the selected pathogen done before in the discussion section and compare with your outputs.

Reviewer #2: 1. “Reverse vaccination”, which is used throughout the manuscript, is not the correct word and should be written “reverse vaccinology” instead.

2. The manuscript is very poor in terms of grammar.

3. The authors did not explain about the algorithm used in the software by which the analysis was performed.

4. The authors should have provided a rationale for using β-defensin 3 and PADRE as adjuvants in the vaccine construct.

5. The authors should have provided a rationale for using TLR4 as receptor in the molecular docking analysis.

6. In the manuscript, only the selected epitopes are given without mentioning the allele, if an additional file containing all the predicted epitopes along with their alleles should have been provided.

7. In molecular dynamics simulation to investigate the stability and flexibility of the dockedcomplex, RMSD and RMSF plots are needed, which unfortunately the authors did not provide.

8. The discussion section is written in a disorganized manner.

6. PLOS authors have the option to publish the peer review history of their article (what does this mean?). If published, this will include your full peer review and any attached files.

Reviewer #1: **Yes: **Muhammad Waqas

Reviewer #2: No

---

## [Author Response · Author response to Decision Letter 0]

7 Jul 2023

Author Response

Reviewer #1:

The authors have done tremendous work entitled "Design of multiepitope vaccine candidate against Brucella type Ⅳ secretion system（T4SS)" in the field of vaccine development. There is space to improve the manuscript to be published in this prestigious journal. Here are some suggestions which will improve the quality of the manuscript. 

Response: Dear reviewer,thank you for reviewing our research article and providing valuable suggestions for revisions. We take your feedback seriously and have made corresponding revisions based on your suggestions. Here is our response to your proposed modification suggestions:

Comments

1.The abstract should start with the introduction of the bacteria that the authors selected. For a clear understanding, add two introduction lines to the bacteria in the abstract. 

Response: Dear reviewer, thank you for your valuable suggestions.We have made corresponding modifications in the manuscript. We have added two introduction lines to the bacteria in the abstract. "Brucellosis is a common zoonosis, which is caused by Brucella infection, and Brucella often infects livestock, leading to abortion and infertility." This sentence is at the beginning of the abstract.

2.The overall abstract is in the form of methodology. What promising results about the vaccine or epitopes the authors get are lacking in the abstract. The abstract should be modified according to the results and a clear conclusion.

Response:Dear reviewer, in the abstract,we have revised the results and conclusions in line with the valuable comments you have made.The following is what we have modified:"At present, human brucellosis remains one of the major public health problems in China. According to previous research, most areas in northwest China, including Xinjiang, Tibet, and other regions, are severely affected by brucella. Although there are vaccines against animal Brucellosis, the effect is often poor. In addition, there is no corresponding vaccine for human Brucellosis infection. Therefore, a new strategy for early prevention and treatment of Brucella is needed. A multi-epitope vaccine should be developed. In this study, we identified the antigenic epitopes of the Brucella type Ⅳ secretion system VirB8 and Virb10 using immunoinformatics approach, and screened out 2 cytotoxic T lymphocyte (CTL) epitopes, 9 helper T lymphocyte (HTL) epitopes, 6 linear B cell epitopes, and 6 conformational B cell epitopes. These advantageous epitopes are spliced together through different linkers to construct a multi-epitope vaccine. The silico tests showed that the multi-epitope vaccine was non-allergenic and strong interaction with TLR4 molecular docking. In immune simulation results, the vaccine construct may be useful in helping brucellosis patients to initiate cellular and humoral immunity. Overall, our findings indicated that the multi-epitope vaccine construct has a high-quality structure and suitable characteristics, which may provide a theoretical basis for the development of a Brucella vaccine."

3.The aim of the study should be discussed, unlike the methodology in the introduction. For better understanding, provide a clear study goal of why the vaccine is essential for the selected bacteria. 

Response:Dear reviewer, in the introductory section, we provide clear study objectives and detail the importance of the vaccine for Brucella based on your valuable suggestions. Our modifications are as follows: "However, there are still many clinical challenges in the diagnosis of brucellosis because of its non-specified clinical features, the slow growth rate in blood cultures, and the complexity of its serodiagnosis[9]. In addition, the disease is easy to develop into chronic, which can affect multiple organs at the same time. Therefore, developing a new approach to early prevention of brucellosis is necessary.Vaccines are an ideal way to prevent brucellosis. Unfortunately, there is still no available human brucellosis vaccine. All commercially available animal vaccines are based on live attenuated strains of Brucella that can induce abortions in pregnant animals and are potentially infectious to humans. Therefore, An efficient Brucella vaccine is urgently needed."

4.In the "1.1 Material sources" section, the authors retrieve the sequences from the UniProt database, but no ID is mentioned of any of the sequences, and also provide the UniProt database link.

Response:Dear reviewer, Due to an oversight on our part we have missed the ID number of the sequence and would very much appreciate the editor's valuable suggestions. Therefore, in the "1.1 Material sources" section, we have added the IDs of the corresponding sequences.

5.In the section "1.2.1 Selection of target proteins", the authors find the antigenicity only for the selected proteins. First, discuss which proteins are selected from the pathogen and calculate the allergenicity of the proteins. 

Response:Dear reviewer,thank you for your valuable suggestions. In the article, we discussed the selected proteins among pathogens. Due to our analysis of the allergenicity of the selected protein and MEV through the online website AllergenFP v.1.0, and the website's analysis results only show whether it is an allergen.The allergenicity result of VirB8 is PROBABLE ALLERGEN. Nevertheless, we finally chose the protein and selected T and B cell dominant epitopes with good Antigenicity and non allergenicity from it. Because by consulting articles written by other authors, this protein has also been used for candidate vaccine design. Finally, the allergenicity results of VirB10 and MEV were PROBABLE NON ALLERGEN. If necessary, pictures can be provided as a basis.

6.The sequence search and material sources sections should be combined in one subtopic and presented at the start of the methodology.

Response:Dear reviewer, thank you for your valuable suggestions. The sequence search and material sources sections have been merged into one sub theme"1.1 Material sources".

7.In the "Prediction of protein T-cell epitopes" section, the authors took six alleles for CTL prediction and three alleles for HTL prediction. Provide the population coverage analysis of these alleles to justify the selection of these alleles.

Response:Dear reviewer, thank you for your valuable suggestions. In the "Prediction of Protein T Cell Epitopes" section, we selected alleles through a reference literature that analyzed alleles in the Chinese Uyghur population. (Shen CM, Zhu BF, Deng YJ, et al. Allele polymorphism and haplotype diversity of HLA-A, -B and -DRB1 loci in sequence-based typing for Chinese Uyghur ethnic group. PLoS One. 2010;5(11):e13458.Published 2010 Nov 4.doi:10.1371/journal.pone. 0013458). The reference number in the article is 29

8.Explain how the final epitopes were selected from the prediction in the "Prediction of protein T-cell epitopes" section.

Response:Dear reviewer, thank you for your valuable suggestions. As described in "Prediction of protein T-cell epitopes",the overlapping sequences of the top 5 with high score, non-allergenic and high immunogenicity from two software were chosen as T cell dominant epitopes. Ultimately, two CTL-dominant epitopes and nine HTL-dominant epitopes were obtained.

9.Similarly, in the B-cell predictions elaborates the process of final epitopes selection from the prediction. 

Response:Dear reviewer, thank you for your valuable suggestions. Similarly, the overlapping sequences of the top 5 with high score, non-allergenic and high immunogenicity were chosen as B cell dominant epitopes. However, for the conformational epitope of B cells, it has no antigenicity and allergenicity, so we finally selected the dominant epitope with the score of 6 above 0.800.

10.The selected epitopes' physiochemical properties, antigenicity and allergenicity, should be calculated and reported as a safe selection for the Multiepitope vaccines. 

Response:Dear reviewer, thank you for your valuable suggestions. We calculated and displayed the physical and chemical properties and Antigenicity of the selected epitopes. The allergenicity analysis results of all selected epitopes are PROBABLE NON ALLERGEN.

11.The authors used the MEV-TLR4 complex in section "1.2.15 Molecular dynamics simulation", but did not mention any docking protocols or details of the Multiepitope vaccine and TLR4 receptor. 

Response:Dear reviewer, thank you for your valuable suggestions. We elaborated on the detailed information of MEV-TLR4 molecular docking and the calculation process of the server in "1.2.10 Molecular Docking".We use HDOCK server for analysis docking, and in the operation page, we submit the protein FASTA formats of TLR4 and MEV. The second step is sequence similarity search. Given a sequence from input or structural transformation, perform a sequence similarity search on the PDB sequence database to find homologous sequences of receptor and ligand molecules. During template selection, if the differences in sequence coverage, similarity, and resolution between the two templates are within 10%, the template in the complex also takes precedence over the others. With the selected templates, the models are produced with the aid of MODELLER. Finally, traditional global docking, a layered docking program based on FFT, is used for assuming the binding direction of global sampling. Finally, we selected the advantage model from the top ten models.

12.The section "1.2.15 Molecular dynamics simulation" is very poorly explained. The simulation protocols used by the server should be explained in detail, like which forcefields are used and what other parameters are provided to the simulation.

Response:Dear reviewer, thank you for your valuable suggestions. Due to the merger of previous subheadings, "1.2.15 Molecular Dynamics Simulation" have been changed to "1.2.14 Molecular Dynamics Simulation". In the iMODS server, since the definition of dihedral angle for backbone atoms N, CAand C is mandatory, we submit the PDB file format of the MEV-TLR4 complex (including atomic coordinates). Next, we selected the CA model from three coarse-grained (CG) models. Becaues it can show that Cα atoms accounting for whole residue mass. Then, we chose JAVA in the JSmol plugin, which is the fastest and most memory-effective mode. Finally,the other parameters remain unchanged. It is worth noting that all indicators were conducted in Normal Mode Analysis (NMA).

13.The authors stated in lines 210-211 that "the deformability, stiffness and stability of the complex with the output results". What analysis they performed after the simulation for what purpose should be adequately explained, like RMSD, RMSF, Radius of gyration, hydrogen bond analysis, PCA, etc.

Response:Dear reviewer, thank you for your valuable suggestions. We use iMODS online analysis software for molecular dynamics simulation of vaccines. There are six graphs in the result output, which are:covariance matrix,elastic network model,B-factor graph,deformability graph,Eigenvalue of the MEV–TLR4 complex and the variance associated to the modes.The graph of deformability peaks represents the construct’s deformable loci where it presents the coil-shaped amino acids. The B-factor graph depicts the complex’s link between the Normal Mode Analysis and PDB regions. The characteristic value is also one of the stability indicators of the composite. A covariance matrix depicts the relationships among amino acid duplets in the dynamical area.The model of an elastic network is represented to order the pair of atoms linked with the springs. The grey color represents the stiffer region. Thus, the results indicate that the vaccine construct was found be stiffer and more stable.We have referred to the research findings of other researchers and found that in molecular dynamics simulations, they did not provide these indicators as you mentioned. We believe that the results of iMODS can also explain the stability of the complex.

14.The figure captions should be with the figure to explain it adequately, not in the manuscript's text. 

Response:Dear reviewer, thank you for your suggestion. We have deleted the figure captions in the manuscript and resubmit figures with the figure captions.

15.The section "2.10 Quality assessment of models" should be appropriately discussed. Provide all the details of how much the favoured region is, allowed and disallowed regions of the model in the text. 

Response:Dear reviewer, thank you for your suggestion. We have discussed the output results in The section "2.10 Quality assessment of models".: Ramachandran plot analysis showing 90.47% in favored, 7.10% in allowed, and 2.43% in disallowed regions of protein residues.We will re elaborate on the detailed information in "2.9 Quality assessment of models". In addition, these outputs are illustrated in the figure captions.

16.The section "2.11 Molecular docking" is very poorly explained. Explain how many hydrogen bonds are formed between the MEV and TLR4 and how many salt bridges are formed. 

Response:Dear reviewer, thank you for your suggestion. We have supplemented this content in "2.10 Molecular Docking". The Ligplot predicts a two-dimensional interaction interface. There are 2 salt bridges with the red dotted line and 4 hydrogen bonds with the green dotted line in this two-dimensional interface. After making modifications, we found that this section is more complete.

17.The authors select the monomer of the TLR4 receptor, but its active form is the dimmer. Is there any specific reason for the monomer selection? Explain it, please, in the methodology.

Response:Dear reviewer, thank you for your suggestion. As is well known, TLR4 is a dimer and is composed of two identical monomers. But when we conducted visual analysis of the MEV-TLR4 complex, we found that, The intermolecular forces between the "B chain" in TLR4 and MEV are more powerful, while the other "A chain" has no intermolecular forces. Therefore, we only show the interaction between one monomer "B chain" in TLR4 and MEV in the visualization diagram. The other monomer is hidden through visualization tools. In addition, other analyses of MEV-TLR4 complexes show that TLR4 interacts with MEV in the form of dimers, ensuring the activity of TLR4.

18.The authors stated in lines 397-399 that "The eigenvalue in the eigenvalue plot is 9.353365e-05, which, according to previous studies, indicates that the complex has low deformability and good stability" but did not provide any sufficient references to the sentence.

Response:Dear reviewer, thank you for your suggestion. Due to our carelessness, using only eigenvalues to represent the structural characteristics of the complex is too arbitrary. Therefore, we have decided to change it to"9.353365e-05 was the calculated eigenvalue for the complicated structure, and it grew steadily in each mode[42]." In the end, the low deformability in the original sentence was removed and the stability was determined by all the results together. We have also provided reference on the revised sentences.

19.The selection "2.15 Molecular dynamics simulation" results are discussed very poorly. Kindly search for some literature on how to explain these results in a better way. The output of the results is explained like a methodology. Focus on what good results you obtained from the output and discuss those just. 

Response:Dear reviewer, thank you for your suggestion. We have discussed the section on "2.14 Molecular Dynamics Simulation" and made changes to the previous conclusion. We elaborated on the meaning of each value in the graph and explained the meaning represented by each graph. Finally, we found that all the results were as expected, our complex had good stability."Additionally, using counting normal mode analysis (NMA) via server iMODS, the stiffness of the mobility and deformability of the complex residues were examined. Less mobility was shown in the complex's deformability plot for the residues and the complex's B-factor was less deviated by the NMA under analysis(Figs 12A, C). Individual (purple) and cumulative (green) variations are depicted as colored bars. The individual variance of each successive mode exhibited a modest decline, according to the variance plot analysis(Fig 12B). 9.353365e-05 was the calculated eigenvalue for the complicated structure and it grew steadily in each mode[42](Fig 12D). The covariance matrix of the complex molecule represents the degree of correlation between different pairs of atomic motions, which can be either correlated (red), anti-correlated (blue), or uncorrelated (white)(Fig 12E). Additionally, an elastic network model of MEV was generated to differentiate the atom pairs connected by springs. In the diagram, each dot represents a spring between the corresponding atom pairs, and, the color of the dot indicates the stiffness of the spring. The darker gray color indicates that the springs are stiffer(Fig 12F)."

20.Add vaccine development studies for the selected pathogen done before in the discussion section and compare with your outputs. 

Response:Dear reviewer, thank you for your suggestion. before the "Discussion" section, we summarized previous research and compared it with our own."At present, there is still no good vaccine available for people suffering from brucellosis. Therefore, more and more people are designing vaccines targeting the outer membrane proteins of Brucella, such as Omp22, Omp19, Omp28, Omp10, Omp25, Omp31and BtpB. However, most vaccine designs are aimed at destroying the Brucella cell membrane and thereby killing bacteria[44,45,46].It is worth noting that this study designed a vaccine for Brucella T4SS in order to disrupt the basic structure of T4SS and cause the bacteria to lose the conditions for intracellular reproduction. Therefore, it blocks the further infection of bacteria in the patient's body, adding a new protective pathway to protect the patient from the persistent infection of Brucella[47]."

Reviewer #2: 

1.“Reverse vaccination”, which is used throughout the manuscript, is not the correct word and should be written “reverse vaccinology” instead.

Response:Dear reviewer,thank you for reviewing our research article and providing valuable suggestions for revisions. We have modified “Reverse vaccination”to  “reverse vaccinology”

2.The manuscript is very poor in terms of grammar.

Response:Dear reviewer, thank you for your suggestion. We have made grammar modifications to the article.

3.The authors did not explain about the algorithm used in the software by which the analysis was performed.

Response:Dear reviewer, thank you for your suggestion. We have reinterpreted the algorithm and running steps of the online analysis software in the manuscript.

4.The authors should have provided a rationale for using β-defensin 3 and PADRE as adjuvants in the vaccine construct.

Response:Dear reviewer, thank you for your suggestion. We explained in the "Discussion" section why we used β- Defensin 3 and PADRE as adjuvants.

5. The authors should have provided a rationale for using TLR4 as receptor in the molecular docking analysis.

Response:Dear reviewer, thank you for your suggestion. we have discussed the selection of TLR4 in the "Introduction" section."The vaccination must work appropriately with the host's immunological receptors in order to trigger the immune response from the host. Therefore, molecular docking was used to check the vaccine construct models' ability to bind to host immunological receptors. TLR4 is the main receptor of LPS and can be activated by LPS. Subsequently, the activity of the MEV-TLR4 complex is also activated, and the immune response of the body is initiated".

6. In the manuscript, only the selected epitopes are given without mentioning the allele, if an additional file containing all the predicted epitopes along with their alleles should have been provided.

Response:Dear reviewer, thank you for your suggestion. We will resubmit all predicted epitopes and alleles in the form of supplementary documents.

7. In molecular dynamics simulation to investigate the stability and flexibility of the dockedcomplex, RMSD and RMSF plots are needed, which unfortunately the authors did not provide.

Response:Dear reviewer, thank you for your suggestion. The iMODS server executed dynamics simulation analysis for the TLR4-vaccine complex structures to verify the mobility and stability of the vaccine’s complex structures and molecules. In addition, we referred to articles from other researchers and did not provide RMSD and RMSF diagrams. Therefore, there are also no RMSD and RMSF plots in this study. (Naveed M, Jabeen K, Naz R, et al. Regulation of Host Immune Response against Enterobacter cloacae Proteins via Computational mRNA Vaccine Design through Transcriptional Modification. Microorganisms. 2022;10(8):1621. Published 2022 Aug 10. doi:10.3390/microorganisms10081621).

8. The discussion section is written in a disorganized manner.

Response:Dear reviewer, thank you for your suggestion. We have provided a more structured and detailed discussion of the 'Discussion' section.

---

## [Editor Report · Decision Letter 1]

11 Jul 2023

PONE-D-23-13551R1Design of multi-epitope vaccine candidate against Brucella type Ⅳ secretion system（T4SS）PLOS ONE

Dear Dr. Zhang,

Thank you for submitting your manuscript to PLOS ONE. After careful consideration, we feel that it has merit but does not fully meet PLOS ONE’s publication criteria as it currently stands. Therefore, we invite you to submit a revised version of the manuscript that addresses the points raised during the review process.

We look forward to receiving your revised manuscript.

Kind regards,

Sheikh Arslan Sehgal, PhD

Academic Editor

PLOS ONE

Journal Requirements:

Additional Editor Comments:

The figures are very poor. Please revise all the figures and improve the resolution and quality of the figure.

---

## [Author Response · Author response to Decision Letter 1]

11 Jul 2023

Author Response

Reviewer #1:

The authors have done tremendous work entitled "Design of multiepitope vaccine candidate against Brucella type Ⅳ secretion system（T4SS)" in the field of vaccine development. There is space to improve the manuscript to be published in this prestigious journal. Here are some suggestions which will improve the quality of the manuscript. 

Response: Dear reviewer,thank you for reviewing our research article and providing valuable suggestions for revisions. We take your feedback seriously and have made corresponding revisions based on your suggestions. Here is our response to your proposed modification suggestions:

Comments

1.The abstract should start with the introduction of the bacteria that the authors selected. For a clear understanding, add two introduction lines to the bacteria in the abstract. 

Response: Dear reviewer, thank you for your valuable suggestions.We have made corresponding modifications in the manuscript. We have added two introduction lines to the bacteria in the abstract. "Brucellosis is a common zoonosis, which is caused by Brucella infection, and Brucella often infects livestock, leading to abortion and infertility." This sentence is at the beginning of the abstract.

2.The overall abstract is in the form of methodology. What promising results about the vaccine or epitopes the authors get are lacking in the abstract. The abstract should be modified according to the results and a clear conclusion.

Response:Dear reviewer, in the abstract,we have revised the results and conclusions in line with the valuable comments you have made.The following is what we have modified:"At present, human brucellosis remains one of the major public health problems in China. According to previous research, most areas in northwest China, including Xinjiang, Tibet, and other regions, are severely affected by brucella. Although there are vaccines against animal Brucellosis, the effect is often poor. In addition, there is no corresponding vaccine for human Brucellosis infection. Therefore, a new strategy for early prevention and treatment of Brucella is needed. A multi-epitope vaccine should be developed. In this study, we identified the antigenic epitopes of the Brucella type Ⅳ secretion system VirB8 and Virb10 using immunoinformatics approach, and screened out 2 cytotoxic T lymphocyte (CTL) epitopes, 9 helper T lymphocyte (HTL) epitopes, 6 linear B cell epitopes, and 6 conformational B cell epitopes. These advantageous epitopes are spliced together through different linkers to construct a multi-epitope vaccine. The silico tests showed that the multi-epitope vaccine was non-allergenic and strong interaction with TLR4 molecular docking. In immune simulation results, the vaccine construct may be useful in helping brucellosis patients to initiate cellular and humoral immunity. Overall, our findings indicated that the multi-epitope vaccine construct has a high-quality structure and suitable characteristics, which may provide a theoretical basis for the development of a Brucella vaccine."

3.The aim of the study should be discussed, unlike the methodology in the introduction. For better understanding, provide a clear study goal of why the vaccine is essential for the selected bacteria. 

Response:Dear reviewer, in the introductory section, we provide clear study objectives and detail the importance of the vaccine for Brucella based on your valuable suggestions. Our modifications are as follows: "However, there are still many clinical challenges in the diagnosis of brucellosis because of its non-specified clinical features, the slow growth rate in blood cultures, and the complexity of its serodiagnosis[9]. In addition, the disease is easy to develop into chronic, which can affect multiple organs at the same time. Therefore, developing a new approach to early prevention of brucellosis is necessary.Vaccines are an ideal way to prevent brucellosis. Unfortunately, there is still no available human brucellosis vaccine. All commercially available animal vaccines are based on live attenuated strains of Brucella that can induce abortions in pregnant animals and are potentially infectious to humans. Therefore, An efficient Brucella vaccine is urgently needed."

4.In the "1.1 Material sources" section, the authors retrieve the sequences from the UniProt database, but no ID is mentioned of any of the sequences, and also provide the UniProt database link.

Response:Dear reviewer, Due to an oversight on our part we have missed the ID number of the sequence and would very much appreciate the editor's valuable suggestions. Therefore, in the "1.1 Material sources" section, we have added the IDs of the corresponding sequences.

5.In the section "1.2.1 Selection of target proteins", the authors find the antigenicity only for the selected proteins. First, discuss which proteins are selected from the pathogen and calculate the allergenicity of the proteins. 

Response:Dear reviewer,thank you for your valuable suggestions. In the article, we discussed the selected proteins among pathogens. Due to our analysis of the allergenicity of the selected protein and MEV through the online website AllergenFP v.1.0, and the website's analysis results only show whether it is an allergen.The allergenicity result of VirB8 is PROBABLE ALLERGEN. Nevertheless, we finally chose the protein and selected T and B cell dominant epitopes with good Antigenicity and non allergenicity from it. Because by consulting articles written by other authors, this protein has also been used for candidate vaccine design. Finally, the allergenicity results of VirB10 and MEV were PROBABLE NON ALLERGEN. If necessary, pictures can be provided as a basis.

6.The sequence search and material sources sections should be combined in one subtopic and presented at the start of the methodology.

Response:Dear reviewer, thank you for your valuable suggestions. The sequence search and material sources sections have been merged into one sub theme"1.1 Material sources".

7.In the "Prediction of protein T-cell epitopes" section, the authors took six alleles for CTL prediction and three alleles for HTL prediction. Provide the population coverage analysis of these alleles to justify the selection of these alleles.

Response:Dear reviewer, thank you for your valuable suggestions. In the "Prediction of Protein T Cell Epitopes" section, we selected alleles through a reference literature that analyzed alleles in the Chinese Uyghur population. (Shen CM, Zhu BF, Deng YJ, et al. Allele polymorphism and haplotype diversity of HLA-A, -B and -DRB1 loci in sequence-based typing for Chinese Uyghur ethnic group. PLoS One. 2010;5(11):e13458.Published 2010 Nov 4.doi:10.1371/journal.pone. 0013458). The reference number in the article is 29

8.Explain how the final epitopes were selected from the prediction in the "Prediction of protein T-cell epitopes" section.

Response:Dear reviewer, thank you for your valuable suggestions. As described in "Prediction of protein T-cell epitopes",the overlapping sequences of the top 5 with high score, non-allergenic and high immunogenicity from two software were chosen as T cell dominant epitopes. Ultimately, two CTL-dominant epitopes and nine HTL-dominant epitopes were obtained.

9.Similarly, in the B-cell predictions elaborates the process of final epitopes selection from the prediction. 

Response:Dear reviewer, thank you for your valuable suggestions. Similarly, the overlapping sequences of the top 5 with high score, non-allergenic and high immunogenicity were chosen as B cell dominant epitopes. However, for the conformational epitope of B cells, it has no antigenicity and allergenicity, so we finally selected the dominant epitope with the score of 6 above 0.800.

10.The selected epitopes' physiochemical properties, antigenicity and allergenicity, should be calculated and reported as a safe selection for the Multiepitope vaccines. 

Response:Dear reviewer, thank you for your valuable suggestions. We calculated and displayed the physical and chemical properties and Antigenicity of the selected epitopes. The allergenicity analysis results of all selected epitopes are PROBABLE NON ALLERGEN.

11.The authors used the MEV-TLR4 complex in section "1.2.15 Molecular dynamics simulation", but did not mention any docking protocols or details of the Multiepitope vaccine and TLR4 receptor. 

Response:Dear reviewer, thank you for your valuable suggestions. We elaborated on the detailed information of MEV-TLR4 molecular docking and the calculation process of the server in "1.2.10 Molecular Docking".We use HDOCK server for analysis docking, and in the operation page, we submit the protein FASTA formats of TLR4 and MEV. The second step is sequence similarity search. Given a sequence from input or structural transformation, perform a sequence similarity search on the PDB sequence database to find homologous sequences of receptor and ligand molecules. During template selection, if the differences in sequence coverage, similarity, and resolution between the two templates are within 10%, the template in the complex also takes precedence over the others. With the selected templates, the models are produced with the aid of MODELLER. Finally, traditional global docking, a layered docking program based on FFT, is used for assuming the binding direction of global sampling. Finally, we selected the advantage model from the top ten models.

12.The section "1.2.15 Molecular dynamics simulation" is very poorly explained. The simulation protocols used by the server should be explained in detail, like which forcefields are used and what other parameters are provided to the simulation.

Response:Dear reviewer, thank you for your valuable suggestions. Due to the merger of previous subheadings, "1.2.15 Molecular Dynamics Simulation" have been changed to "1.2.14 Molecular Dynamics Simulation". In the iMODS server, since the definition of dihedral angle for backbone atoms N, CAand C is mandatory, we submit the PDB file format of the MEV-TLR4 complex (including atomic coordinates). Next, we selected the CA model from three coarse-grained (CG) models. Becaues it can show that Cα atoms accounting for whole residue mass. Then, we chose JAVA in the JSmol plugin, which is the fastest and most memory-effective mode. Finally,the other parameters remain unchanged. It is worth noting that all indicators were conducted in Normal Mode Analysis (NMA).

13.The authors stated in lines 210-211 that "the deformability, stiffness and stability of the complex with the output results". What analysis they performed after the simulation for what purpose should be adequately explained, like RMSD, RMSF, Radius of gyration, hydrogen bond analysis, PCA, etc.

Response:Dear reviewer, thank you for your valuable suggestions. We use iMODS online analysis software for molecular dynamics simulation of vaccines. There are six graphs in the result output, which are:covariance matrix,elastic network model,B-factor graph,deformability graph,Eigenvalue of the MEV–TLR4 complex and the variance associated to the modes.The graph of deformability peaks represents the construct’s deformable loci where it presents the coil-shaped amino acids. The B-factor graph depicts the complex’s link between the Normal Mode Analysis and PDB regions. The characteristic value is also one of the stability indicators of the composite. A covariance matrix depicts the relationships among amino acid duplets in the dynamical area.The model of an elastic network is represented to order the pair of atoms linked with the springs. The grey color represents the stiffer region. Thus, the results indicate that the vaccine construct was found be stiffer and more stable.We have referred to the research findings of other researchers and found that in molecular dynamics simulations, they did not provide these indicators as you mentioned. We believe that the results of iMODS can also explain the stability of the complex.

14.The figure captions should be with the figure to explain it adequately, not in the manuscript's text. 

Response:Dear reviewer, thank you for your suggestion. We have deleted the figure captions in the manuscript and resubmit figures with the figure captions.

15.The section "2.10 Quality assessment of models" should be appropriately discussed. Provide all the details of how much the favoured region is, allowed and disallowed regions of the model in the text. 

Response:Dear reviewer, thank you for your suggestion. We have discussed the output results in The section "2.10 Quality assessment of models".: Ramachandran plot analysis showing 90.47% in favored, 7.10% in allowed, and 2.43% in disallowed regions of protein residues.We will re elaborate on the detailed information in "2.9 Quality assessment of models". In addition, these outputs are illustrated in the figure captions.

16.The section "2.11 Molecular docking" is very poorly explained. Explain how many hydrogen bonds are formed between the MEV and TLR4 and how many salt bridges are formed. 

Response:Dear reviewer, thank you for your suggestion. We have supplemented this content in "2.10 Molecular Docking". The Ligplot predicts a two-dimensional interaction interface. There are 2 salt bridges with the red dotted line and 4 hydrogen bonds with the green dotted line in this two-dimensional interface. After making modifications, we found that this section is more complete.

17.The authors select the monomer of the TLR4 receptor, but its active form is the dimmer. Is there any specific reason for the monomer selection? Explain it, please, in the methodology.

Response:Dear reviewer, thank you for your suggestion. As is well known, TLR4 is a dimer and is composed of two identical monomers. But when we conducted visual analysis of the MEV-TLR4 complex, we found that, The intermolecular forces between the "B chain" in TLR4 and MEV are more powerful, while the other "A chain" has no intermolecular forces. Therefore, we only show the interaction between one monomer "B chain" in TLR4 and MEV in the visualization diagram. The other monomer is hidden through visualization tools. In addition, other analyses of MEV-TLR4 complexes show that TLR4 interacts with MEV in the form of dimers, ensuring the activity of TLR4.

18.The authors stated in lines 397-399 that "The eigenvalue in the eigenvalue plot is 9.353365e-05, which, according to previous studies, indicates that the complex has low deformability and good stability" but did not provide any sufficient references to the sentence.

Response:Dear reviewer, thank you for your suggestion. Due to our carelessness, using only eigenvalues to represent the structural characteristics of the complex is too arbitrary. Therefore, we have decided to change it to"9.353365e-05 was the calculated eigenvalue for the complicated structure, and it grew steadily in each mode[42]." In the end, the low deformability in the original sentence was removed and the stability was determined by all the results together. We have also provided reference on the revised sentences.

19.The selection "2.15 Molecular dynamics simulation" results are discussed very poorly. Kindly search for some literature on how to explain these results in a better way. The output of the results is explained like a methodology. Focus on what good results you obtained from the output and discuss those just. 

Response:Dear reviewer, thank you for your suggestion. We have discussed the section on "2.14 Molecular Dynamics Simulation" and made changes to the previous conclusion. We elaborated on the meaning of each value in the graph and explained the meaning represented by each graph. Finally, we found that all the results were as expected, our complex had good stability."Additionally, using counting normal mode analysis (NMA) via server iMODS, the stiffness of the mobility and deformability of the complex residues were examined. Less mobility was shown in the complex's deformability plot for the residues and the complex's B-factor was less deviated by the NMA under analysis(Figs 12A, C). Individual (purple) and cumulative (green) variations are depicted as colored bars. The individual variance of each successive mode exhibited a modest decline, according to the variance plot analysis(Fig 12B). 9.353365e-05 was the calculated eigenvalue for the complicated structure and it grew steadily in each mode[42](Fig 12D). The covariance matrix of the complex molecule represents the degree of correlation between different pairs of atomic motions, which can be either correlated (red), anti-correlated (blue), or uncorrelated (white)(Fig 12E). Additionally, an elastic network model of MEV was generated to differentiate the atom pairs connected by springs. In the diagram, each dot represents a spring between the corresponding atom pairs, and, the color of the dot indicates the stiffness of the spring. The darker gray color indicates that the springs are stiffer(Fig 12F)."

20.Add vaccine development studies for the selected pathogen done before in the discussion section and compare with your outputs. 

Response:Dear reviewer, thank you for your suggestion. before the "Discussion" section, we summarized previous research and compared it with our own."At present, there is still no good vaccine available for people suffering from brucellosis. Therefore, more and more people are designing vaccines targeting the outer membrane proteins of Brucella, such as Omp22, Omp19, Omp28, Omp10, Omp25, Omp31and BtpB. However, most vaccine designs are aimed at destroying the Brucella cell membrane and thereby killing bacteria[44,45,46].It is worth noting that this study designed a vaccine for Brucella T4SS in order to disrupt the basic structure of T4SS and cause the bacteria to lose the conditions for intracellular reproduction. Therefore, it blocks the further infection of bacteria in the patient's body, adding a new protective pathway to protect the patient from the persistent infection of Brucella[47]."

Reviewer #2: 

1.“Reverse vaccination”, which is used throughout the manuscript, is not the correct word and should be written “reverse vaccinology” instead.

Response:Dear reviewer,thank you for reviewing our research article and providing valuable suggestions for revisions. We have modified “Reverse vaccination”to  “reverse vaccinology”

2.The manuscript is very poor in terms of grammar.

Response:Dear reviewer, thank you for your suggestion. We have made grammar modifications to the article.

3.The authors did not explain about the algorithm used in the software by which the analysis was performed.

Response:Dear reviewer, thank you for your suggestion. We have reinterpreted the algorithm and running steps of the online analysis software in the manuscript.

4.The authors should have provided a rationale for using β-defensin 3 and PADRE as adjuvants in the vaccine construct.

Response:Dear reviewer, thank you for your suggestion. We explained in the "Discussion" section why we used β- Defensin 3 and PADRE as adjuvants.

5. The authors should have provided a rationale for using TLR4 as receptor in the molecular docking analysis.

Response:Dear reviewer, thank you for your suggestion. we have discussed the selection of TLR4 in the "Introduction" section."The vaccination must work appropriately with the host's immunological receptors in order to trigger the immune response from the host. Therefore, molecular docking was used to check the vaccine construct models' ability to bind to host immunological receptors. TLR4 is the main receptor of LPS and can be activated by LPS. Subsequently, the activity of the MEV-TLR4 complex is also activated, and the immune response of the body is initiated".

6. In the manuscript, only the selected epitopes are given without mentioning the allele, if an additional file containing all the predicted epitopes along with their alleles should have been provided.

Response:Dear reviewer, thank you for your suggestion. We will resubmit all predicted epitopes and alleles in the form of supplementary documents.

7. In molecular dynamics simulation to investigate the stability and flexibility of the dockedcomplex, RMSD and RMSF plots are needed, which unfortunately the authors did not provide.

Response:Dear reviewer, thank you for your suggestion. The iMODS server executed dynamics simulation analysis for the TLR4-vaccine complex structures to verify the mobility and stability of the vaccine’s complex structures and molecules. In addition, we referred to articles from other researchers and did not provide RMSD and RMSF diagrams. Therefore, there are also no RMSD and RMSF plots in this study. (Naveed M, Jabeen K, Naz R, et al. Regulation of Host Immune Response against Enterobacter cloacae Proteins via Computational mRNA Vaccine Design through Transcriptional Modification. Microorganisms. 2022;10(8):1621. Published 2022 Aug 10. doi:10.3390/microorganisms10081621).

8. The discussion section is written in a disorganized manner.

Response:Dear reviewer, thank you for your suggestion. We have provided a more structured and detailed discussion of the 'Discussion' section.

---

## [Editor Report · Decision Letter 2]

13 Jul 2023

PONE-D-23-13551R2Design of multi-epitope vaccine candidate against Brucella type Ⅳ secretion system（T4SS）PLOS ONE

Dear Dr. Zhang,

Thank you for submitting your manuscript to PLOS ONE. After careful consideration, we feel that it has merit but does not fully meet PLOS ONE’s publication criteria as it currently stands. Therefore, we invite you to submit a revised version of the manuscript that addresses the points raised during the review process.

We look forward to receiving your revised manuscript.

Kind regards,

Sheikh Arslan Sehgal, PhD

Academic Editor

PLOS ONE

Journal Requirements:

Additional Editor Comments:

Figures are very poor and unable to read. Improve the quality and resolution of the figures.

---

## [Author Response · Author response to Decision Letter 2]

19 Jul 2023

Author Response

Reviewer #1:

The authors have done tremendous work entitled "Design of multiepitope vaccine candidate against Brucella type Ⅳ secretion system（T4SS)" in the field of vaccine development. There is space to improve the manuscript to be published in this prestigious journal. Here are some suggestions which will improve the quality of the manuscript. 

Response: Dear reviewer,thank you for reviewing our research article and providing valuable suggestions for revisions. We take your feedback seriously and have made corresponding revisions based on your suggestions. Here is our response to your proposed modification suggestions:

Comments

1.The abstract should start with the introduction of the bacteria that the authors selected. For a clear understanding, add two introduction lines to the bacteria in the abstract. 

Response: Dear reviewer, thank you for your valuable suggestions.We have made corresponding modifications in the manuscript. We have added two introduction lines to the bacteria in the abstract. "Brucellosis is a common zoonosis, which is caused by Brucella infection, and Brucella often infects livestock, leading to abortion and infertility." This sentence is at the beginning of the abstract.

2.The overall abstract is in the form of methodology. What promising results about the vaccine or epitopes the authors get are lacking in the abstract. The abstract should be modified according to the results and a clear conclusion.

Response:Dear reviewer, in the abstract,we have revised the results and conclusions in line with the valuable comments you have made.The following is what we have modified:"At present, human brucellosis remains one of the major public health problems in China. According to previous research, most areas in northwest China, including Xinjiang, Tibet, and other regions, are severely affected by brucella. Although there are vaccines against animal Brucellosis, the effect is often poor. In addition, there is no corresponding vaccine for human Brucellosis infection. Therefore, a new strategy for early prevention and treatment of Brucella is needed. A multi-epitope vaccine should be developed. In this study, we identified the antigenic epitopes of the Brucella type Ⅳ secretion system VirB8 and Virb10 using immunoinformatics approach, and screened out 2 cytotoxic T lymphocyte (CTL) epitopes, 9 helper T lymphocyte (HTL) epitopes, 6 linear B cell epitopes, and 6 conformational B cell epitopes. These advantageous epitopes are spliced together through different linkers to construct a multi-epitope vaccine. The silico tests showed that the multi-epitope vaccine was non-allergenic and strong interaction with TLR4 molecular docking. In immune simulation results, the vaccine construct may be useful in helping brucellosis patients to initiate cellular and humoral immunity. Overall, our findings indicated that the multi-epitope vaccine construct has a high-quality structure and suitable characteristics, which may provide a theoretical basis for the development of a Brucella vaccine."

3.The aim of the study should be discussed, unlike the methodology in the introduction. For better understanding, provide a clear study goal of why the vaccine is essential for the selected bacteria. 

Response:Dear reviewer, in the introductory section, we provide clear study objectives and detail the importance of the vaccine for Brucella based on your valuable suggestions. Our modifications are as follows: "However, there are still many clinical challenges in the diagnosis of brucellosis because of its non-specified clinical features, the slow growth rate in blood cultures, and the complexity of its serodiagnosis[9]. In addition, the disease is easy to develop into chronic, which can affect multiple organs at the same time. Therefore, developing a new approach to early prevention of brucellosis is necessary.Vaccines are an ideal way to prevent brucellosis. Unfortunately, there is still no available human brucellosis vaccine. All commercially available animal vaccines are based on live attenuated strains of Brucella that can induce abortions in pregnant animals and are potentially infectious to humans. Therefore, An efficient Brucella vaccine is urgently needed."

4.In the "1.1 Material sources" section, the authors retrieve the sequences from the UniProt database, but no ID is mentioned of any of the sequences, and also provide the UniProt database link.

Response:Dear reviewer, Due to an oversight on our part we have missed the ID number of the sequence and would very much appreciate the editor's valuable suggestions. Therefore, in the "1.1 Material sources" section, we have added the IDs of the corresponding sequences.

5.In the section "1.2.1 Selection of target proteins", the authors find the antigenicity only for the selected proteins. First, discuss which proteins are selected from the pathogen and calculate the allergenicity of the proteins. 

Response:Dear reviewer,thank you for your valuable suggestions. In the article, we discussed the selected proteins among pathogens. Due to our analysis of the allergenicity of the selected protein and MEV through the online website AllergenFP v.1.0, and the website's analysis results only show whether it is an allergen.The allergenicity result of VirB8 is PROBABLE ALLERGEN. Nevertheless, we finally chose the protein and selected T and B cell dominant epitopes with good Antigenicity and non allergenicity from it. Because by consulting articles written by other authors, this protein has also been used for candidate vaccine design. Finally, the allergenicity results of VirB10 and MEV were PROBABLE NON ALLERGEN. If necessary, pictures can be provided as a basis.

6.The sequence search and material sources sections should be combined in one subtopic and presented at the start of the methodology.

Response:Dear reviewer, thank you for your valuable suggestions. The sequence search and material sources sections have been merged into one sub theme"1.1 Material sources".

7.In the "Prediction of protein T-cell epitopes" section, the authors took six alleles for CTL prediction and three alleles for HTL prediction. Provide the population coverage analysis of these alleles to justify the selection of these alleles.

Response:Dear reviewer, thank you for your valuable suggestions. In the "Prediction of Protein T Cell Epitopes" section, we selected alleles through a reference literature that analyzed alleles in the Chinese Uyghur population. (Shen CM, Zhu BF, Deng YJ, et al. Allele polymorphism and haplotype diversity of HLA-A, -B and -DRB1 loci in sequence-based typing for Chinese Uyghur ethnic group. PLoS One. 2010;5(11):e13458.Published 2010 Nov 4.doi:10.1371/journal.pone. 0013458). The reference number in the article is 29

8.Explain how the final epitopes were selected from the prediction in the "Prediction of protein T-cell epitopes" section.

Response:Dear reviewer, thank you for your valuable suggestions. As described in "Prediction of protein T-cell epitopes",the overlapping sequences of the top 5 with high score, non-allergenic and high immunogenicity from two software were chosen as T cell dominant epitopes. Ultimately, two CTL-dominant epitopes and nine HTL-dominant epitopes were obtained.

9.Similarly, in the B-cell predictions elaborates the process of final epitopes selection from the prediction. 

Response:Dear reviewer, thank you for your valuable suggestions. Similarly, the overlapping sequences of the top 5 with high score, non-allergenic and high immunogenicity were chosen as B cell dominant epitopes. However, for the conformational epitope of B cells, it has no antigenicity and allergenicity, so we finally selected the dominant epitope with the score of 6 above 0.800.

10.The selected epitopes' physiochemical properties, antigenicity and allergenicity, should be calculated and reported as a safe selection for the Multiepitope vaccines. 

Response:Dear reviewer, thank you for your valuable suggestions. We calculated and displayed the physical and chemical properties and Antigenicity of the selected epitopes. The allergenicity analysis results of all selected epitopes are PROBABLE NON ALLERGEN.

11.The authors used the MEV-TLR4 complex in section "1.2.15 Molecular dynamics simulation", but did not mention any docking protocols or details of the Multiepitope vaccine and TLR4 receptor. 

Response:Dear reviewer, thank you for your valuable suggestions. We elaborated on the detailed information of MEV-TLR4 molecular docking and the calculation process of the server in "1.2.10 Molecular Docking".We use HDOCK server for analysis docking, and in the operation page, we submit the protein FASTA formats of TLR4 and MEV. The second step is sequence similarity search. Given a sequence from input or structural transformation, perform a sequence similarity search on the PDB sequence database to find homologous sequences of receptor and ligand molecules. During template selection, if the differences in sequence coverage, similarity, and resolution between the two templates are within 10%, the template in the complex also takes precedence over the others. With the selected templates, the models are produced with the aid of MODELLER. Finally, traditional global docking, a layered docking program based on FFT, is used for assuming the binding direction of global sampling. Finally, we selected the advantage model from the top ten models.

12.The section "1.2.15 Molecular dynamics simulation" is very poorly explained. The simulation protocols used by the server should be explained in detail, like which forcefields are used and what other parameters are provided to the simulation.

Response:Dear reviewer, thank you for your valuable suggestions. Due to the merger of previous subheadings, "1.2.15 Molecular Dynamics Simulation" have been changed to "1.2.14 Molecular Dynamics Simulation". In the iMODS server, since the definition of dihedral angle for backbone atoms N, CAand C is mandatory, we submit the PDB file format of the MEV-TLR4 complex (including atomic coordinates). Next, we selected the CA model from three coarse-grained (CG) models. Becaues it can show that Cα atoms accounting for whole residue mass. Then, we chose JAVA in the JSmol plugin, which is the fastest and most memory-effective mode. Finally,the other parameters remain unchanged. It is worth noting that all indicators were conducted in Normal Mode Analysis (NMA).

13.The authors stated in lines 210-211 that "the deformability, stiffness and stability of the complex with the output results". What analysis they performed after the simulation for what purpose should be adequately explained, like RMSD, RMSF, Radius of gyration, hydrogen bond analysis, PCA, etc.

Response:Dear reviewer, thank you for your valuable suggestions. We use iMODS online analysis software for molecular dynamics simulation of vaccines. There are six graphs in the result output, which are:covariance matrix,elastic network model,B-factor graph,deformability graph,Eigenvalue of the MEV–TLR4 complex and the variance associated to the modes.The graph of deformability peaks represents the construct’s deformable loci where it presents the coil-shaped amino acids. The B-factor graph depicts the complex’s link between the Normal Mode Analysis and PDB regions. The characteristic value is also one of the stability indicators of the composite. A covariance matrix depicts the relationships among amino acid duplets in the dynamical area.The model of an elastic network is represented to order the pair of atoms linked with the springs. The grey color represents the stiffer region. Thus, the results indicate that the vaccine construct was found be stiffer and more stable.We have referred to the research findings of other researchers and found that in molecular dynamics simulations, they did not provide these indicators as you mentioned. We believe that the results of iMODS can also explain the stability of the complex.

14.The figure captions should be with the figure to explain it adequately, not in the manuscript's text. 

Response:Dear reviewer, thank you for your suggestion. We have deleted the figure captions in the manuscript and resubmit figures with the figure captions.

15.The section "2.10 Quality assessment of models" should be appropriately discussed. Provide all the details of how much the favoured region is, allowed and disallowed regions of the model in the text. 

Response:Dear reviewer, thank you for your suggestion. We have discussed the output results in The section "2.10 Quality assessment of models".: Ramachandran plot analysis showing 90.47% in favored, 7.10% in allowed, and 2.43% in disallowed regions of protein residues.We will re elaborate on the detailed information in "2.9 Quality assessment of models". In addition, these outputs are illustrated in the figure captions.

16.The section "2.11 Molecular docking" is very poorly explained. Explain how many hydrogen bonds are formed between the MEV and TLR4 and how many salt bridges are formed. 

Response:Dear reviewer, thank you for your suggestion. We have supplemented this content in "2.10 Molecular Docking". The Ligplot predicts a two-dimensional interaction interface. There are 2 salt bridges with the red dotted line and 4 hydrogen bonds with the green dotted line in this two-dimensional interface. After making modifications, we found that this section is more complete.

17.The authors select the monomer of the TLR4 receptor, but its active form is the dimmer. Is there any specific reason for the monomer selection? Explain it, please, in the methodology.

Response:Dear reviewer, thank you for your suggestion. As is well known, TLR4 is a dimer and is composed of two identical monomers. But when we conducted visual analysis of the MEV-TLR4 complex, we found that, The intermolecular forces between the "B chain" in TLR4 and MEV are more powerful, while the other "A chain" has no intermolecular forces. Therefore, we only show the interaction between one monomer "B chain" in TLR4 and MEV in the visualization diagram. The other monomer is hidden through visualization tools. In addition, other analyses of MEV-TLR4 complexes show that TLR4 interacts with MEV in the form of dimers, ensuring the activity of TLR4.

18.The authors stated in lines 397-399 that "The eigenvalue in the eigenvalue plot is 9.353365e-05, which, according to previous studies, indicates that the complex has low deformability and good stability" but did not provide any sufficient references to the sentence.

Response:Dear reviewer, thank you for your suggestion. Due to our carelessness, using only eigenvalues to represent the structural characteristics of the complex is too arbitrary. Therefore, we have decided to change it to"9.353365e-05 was the calculated eigenvalue for the complicated structure, and it grew steadily in each mode[42]." In the end, the low deformability in the original sentence was removed and the stability was determined by all the results together. We have also provided reference on the revised sentences.

19.The selection "2.15 Molecular dynamics simulation" results are discussed very poorly. Kindly search for some literature on how to explain these results in a better way. The output of the results is explained like a methodology. Focus on what good results you obtained from the output and discuss those just. 

Response:Dear reviewer, thank you for your suggestion. We have discussed the section on "2.14 Molecular Dynamics Simulation" and made changes to the previous conclusion. We elaborated on the meaning of each value in the graph and explained the meaning represented by each graph. Finally, we found that all the results were as expected, our complex had good stability."Additionally, using counting normal mode analysis (NMA) via server iMODS, the stiffness of the mobility and deformability of the complex residues were examined. Less mobility was shown in the complex's deformability plot for the residues and the complex's B-factor was less deviated by the NMA under analysis(Figs 12A, C). Individual (purple) and cumulative (green) variations are depicted as colored bars. The individual variance of each successive mode exhibited a modest decline, according to the variance plot analysis(Fig 12B). 9.353365e-05 was the calculated eigenvalue for the complicated structure and it grew steadily in each mode[42](Fig 12D). The covariance matrix of the complex molecule represents the degree of correlation between different pairs of atomic motions, which can be either correlated (red), anti-correlated (blue), or uncorrelated (white)(Fig 12E). Additionally, an elastic network model of MEV was generated to differentiate the atom pairs connected by springs. In the diagram, each dot represents a spring between the corresponding atom pairs, and, the color of the dot indicates the stiffness of the spring. The darker gray color indicates that the springs are stiffer(Fig 12F)."

20.Add vaccine development studies for the selected pathogen done before in the discussion section and compare with your outputs. 

Response:Dear reviewer, thank you for your suggestion. before the "Discussion" section, we summarized previous research and compared it with our own."At present, there is still no good vaccine available for people suffering from brucellosis. Therefore, more and more people are designing vaccines targeting the outer membrane proteins of Brucella, such as Omp22, Omp19, Omp28, Omp10, Omp25, Omp31and BtpB. However, most vaccine designs are aimed at destroying the Brucella cell membrane and thereby killing bacteria[44,45,46].It is worth noting that this study designed a vaccine for Brucella T4SS in order to disrupt the basic structure of T4SS and cause the bacteria to lose the conditions for intracellular reproduction. Therefore, it blocks the further infection of bacteria in the patient's body, adding a new protective pathway to protect the patient from the persistent infection of Brucella[47]."

Reviewer #2: 

1.“Reverse vaccination”, which is used throughout the manuscript, is not the correct word and should be written “reverse vaccinology” instead.

Response:Dear reviewer,thank you for reviewing our research article and providing valuable suggestions for revisions. We have modified “Reverse vaccination”to  “reverse vaccinology”

2.The manuscript is very poor in terms of grammar.

Response:Dear reviewer, thank you for your suggestion. We have made grammar modifications to the article.

3.The authors did not explain about the algorithm used in the software by which the analysis was performed.

Response:Dear reviewer, thank you for your suggestion. We have reinterpreted the algorithm and running steps of the online analysis software in the manuscript.

4.The authors should have provided a rationale for using β-defensin 3 and PADRE as adjuvants in the vaccine construct.

Response:Dear reviewer, thank you for your suggestion. We explained in the "Discussion" section why we used β- Defensin 3 and PADRE as adjuvants.

5. The authors should have provided a rationale for using TLR4 as receptor in the molecular docking analysis.

Response:Dear reviewer, thank you for your suggestion. we have discussed the selection of TLR4 in the "Introduction" section."The vaccination must work appropriately with the host's immunological receptors in order to trigger the immune response from the host. Therefore, molecular docking was used to check the vaccine construct models' ability to bind to host immunological receptors. TLR4 is the main receptor of LPS and can be activated by LPS. Subsequently, the activity of the MEV-TLR4 complex is also activated, and the immune response of the body is initiated".

6. In the manuscript, only the selected epitopes are given without mentioning the allele, if an additional file containing all the predicted epitopes along with their alleles should have been provided.

Response:Dear reviewer, thank you for your suggestion. We will resubmit all predicted epitopes and alleles in the form of supplementary documents.

7. In molecular dynamics simulation to investigate the stability and flexibility of the dockedcomplex, RMSD and RMSF plots are needed, which unfortunately the authors did not provide.

Response:Dear reviewer, thank you for your suggestion. The iMODS server executed dynamics simulation analysis for the TLR4-vaccine complex structures to verify the mobility and stability of the vaccine’s complex structures and molecules. In addition, we referred to articles from other researchers and did not provide RMSD and RMSF diagrams. Therefore, there are also no RMSD and RMSF plots in this study. (Naveed M, Jabeen K, Naz R, et al. Regulation of Host Immune Response against Enterobacter cloacae Proteins via Computational mRNA Vaccine Design through Transcriptional Modification. Microorganisms. 2022;10(8):1621. Published 2022 Aug 10. doi:10.3390/microorganisms10081621).

8. The discussion section is written in a disorganized manner.

Response:Dear reviewer, thank you for your suggestion. We have provided a more structured and detailed discussion of the 'Discussion' section.

---

## [Editor Report · Decision Letter 3]

24 Jul 2023

针对布鲁氏菌IV.型分泌系统（T4SS）的多表位候选疫苗设计

PONE-D-23-13551R3

Dear Dr. Zhang,

We’re pleased to inform you that your manuscript has been judged scientifically suitable for publication and will be formally accepted for publication once it meets all outstanding technical requirements.

Kind regards,

Sheikh Arslan Sehgal, PhD

Academic Editor

PLOS ONE
---

## [Editor Report · Acceptance letter]

31 Jul 2023

PONE-D-23-13551R3 

Design of multi-epitope vaccine candidate against Brucella type Ⅳ secretion system（T4SS） 

Dear Dr. Zhang:

I'm pleased to inform you that your manuscript has been deemed suitable for publication in PLOS ONE. Congratulations! Your manuscript is now with our production department. 

Kind regards, 

on behalf of

Dr Sheikh Arslan Sehgal 

Academic Editor

PLOS ONE